



**Enhanced photodegradation of dimethoxybenzene isomers in/on ice compared to in aqueous**
**solution**
Ted Hullar[1], Theo Tran[1], Zekun Chen[2], Fernanda Bononi[2], Oliver Palmer[1,3], Davide Donadio[2], and
Cort Anastasio[1, *]
[1] Department of Land, Air and Water Resources, University of California, Davis, One Shields
Avenue, Davis, CA 95616, USA
[2] Department of Chemistry, University of California, Davis, One Shields Avenue, Davis, CA
95616, USA
[3] Now at TeraPore Technologies, 407 Cabot Road, South San Francisco, CA  94080
[*] Corresponding author, canastasio@ucdavis.edu, (530) 754-6095
**Abstract**
Photochemical reactions of contaminants in snow and ice can be important sources and sinks for
various organic and inorganic compounds. Snow contaminants can be found in the bulk ice
matrix, in internal liquid-like regions (LLRs), or in quasi-liquid layers (QLLs) at the air-ice
interface, where they can readily exchange with the firn air.  Some studies have reported that
direct photochemical reactions occur faster in LLRs and QLLs than in aqueous solution, while
others have found similar rates.  Here, we measure the photodegradation rate constants of the
three dimethoxybenzene isomers under varying experimental conditions, including in aqueous
solution, in LLRs, and at the air-ice interface of nature-identical snow.  Relative to aqueous
solution, we find modest photodegradation enhancements (3- and 6-fold) in LLRs for two of the
isomers, and larger enhancements (15- to 30-fold) at the air-ice interface for all three isomers.
We use computational modeling to assess the impact of light absorbance changes on
photodegradation rate enhancements at the interface.  We find small (2-5 nm) bathochromic (red)
absorbance shifts at the interface relative to in solution, which increases light absorption, but this
factor only accounts for less than 50% of the measured rate constant enhancements.  The major
factor responsible for photodegradation rate enhancements at the air-ice interface appears to be
more efficient photodecay: estimated dimethoxybenzene quantum yields are 6- to 24-fold larger
at the interface compared to in aqueous solution and account for the majority (51-96%) of the
observed enhancements.  Using a hypothetical model compound with an assumed Gaussian-
shaped absorbance peak, we find that a shift in the peak to higher or lower wavelengths can have
a minor to substantial impact on photodecay rate constants, depending on the original location of
the peak and the magnitude of the shift.  Changes in other peak properties at the air-ice interface,
such as peak width and height (i.e., molar absorptivity) can also impact rates of light absorption
and direct photodecay.



**1. Introduction**
Snow and ice contain a wide variety of chemical compounds (Grannas et al. 2006), which can be
transformed via photochemical reactions (Bartels-Rausch et al. 2014, Domine and Shepson 2002,
Grannas et al. 2007).  While snow and ice are comprised primarily of crystalline water ice, under
environmental conditions there are also small areas of disordered water molecules that contain
most of the solutes present in a snowpack (Barret et al. 2011, Bartels-Rausch et al. 2014,
Grannas et al. 2007, Jacobi et al. 2004).  At the air-ice interface, these regions are called quasi-
liquid layers (QLLs), while those located at ice grain boundaries and other locations  within the
ice matrix are referred to as liquid-like regions (LLRs).  Photochemistry can be important in
snowpacks (Grannas et al. 2007), as light can penetrate tens of centimeters below the snow
surface (France et al. 2011, Galbavy et al. 2007, Phillips and Simpson 2005).  Photochemical
reactions are classified as either direct – where a compound absorbs sunlight and is transformed
– or indirect – where a reactive species (e.g., hydroxyl radical) formed from a direct
photoreaction reacts with the compound of interest.
Despite their importance, only a small number of direct photochemical reactions have been
studied in/on ice, with variable and occasionally conflicting findings.  Measurements of direct
photodegradation rates for a number of inorganic solutes (e.g., nitrate, nitrite, and hydrogen
peroxide) found the same temperature dependence in aqueous solution and LLRs, suggesting
both compartments provide similar environments for chemical reactions (Chu and Anastasio
2003, Chu and Anastasio 2005, Chu and Anastasio 2007).  The picture is more complicated for
PAHs (polycyclic aromatic hydrocarbons).  Two studies found little difference in PAH
photochemistry in/on ice compared to solution:  phenanthrene, pyrene, and fluoranthene had
similar photodegradation rates in aqueous solution and in LLRs (Ram and Anastasio 2009),
while anthracene and pyrene had similar rates in aqueous solution, LLRs, and at the air-ice
inteface (QLLs) (Hullar et al. 2018).  However, two other studies reported that the photodecay of
anthracene and naphthalene were faster in LLRs and at the air-ice interface compared to in
solution (2007, Kahan et al. 2010).  Harmine has also been reported to photodegrade faster at the
air-ice interface (Kahan et al. 2010).  Most recently, we found that guaiacol photodegradation
was somewhat faster in LLRs, and considerably faster at the-air ice interface, than in aqueous
solution (Hullar et al. 2020).
To evaluate the possible causes of photodegradation enhancements in/on ice compared to
solution, consider the variables that control the direct photodecay rate for a chemical "C" (M s$^{-1}$)
(Chu and Anastasio 2003):

$$\frac{d[C]}{dt} = - \sum_\lambda \frac{2303}{N_A} I_\lambda \, \Delta\lambda \, \Phi_{C,\lambda} \, \varepsilon_{C,\lambda} \, [C] \qquad (1)$$

where 2303 is a factor for units (1000 cm$^3$ L$^{-1}$) and for converting the measurements from base
10 to base $e$, $N_A$ is Avogadro's number (6.022 × 10$^{23}$ molecules mol$^{-1}$), $I_\lambda$ is the actinic flux at
each wavelength (photons cm$^{-2}$ s$^{-1}$ nm$^{-1}$), $\Delta\lambda$ is the wavelength interval between photon flux data
points (nm), $\varepsilon_{C,\lambda}$ is the molar absorptivity for C (M$^{-1}$ cm$^{-1}$), $\Phi_{C,\lambda}$ is the quantum yield for loss of C
(molecule photon$^{-1}$), and [C] is the concentration (M).  Based on equation 1, three factors could
increase reaction rates in/on ice relative to solution: higher local photon fluxes, a bathochromic





(red) shift in molar absorptivity ($\varepsilon_{C,\lambda}$) towards longer wavelengths, which have greater photon
fluxes, or larger quantum yields.
Many previous studies did not measure photon fluxes at the point of the reaction, so it is difficult
to accurately determine the significance of local flux differences in accounting for photodecay
enhancements in, or on, ice.  However, measurements in different solute locations, e.g., in
solution, in LLRs, and at the air-ice interface, found that photon fluxes varied by less than a
factor of 1.5 (McFall and Anastasio 2016).  In addition, in our recent work with guaiacol we
normalized photodecay rate constants by photon flux but still saw large differences in rate
constants between solution, in ice, and at the air-ice interface (Hullar et al. 2020).  Thus local
photon flux differences do not appear to a major factor in observed reaction rate enhancements
in/on ice.
Because solar photon fluxes increase by several orders of magnitude between 295 and 400 nm
(Madronich and Flocke 1998), even a small shift in compound absorbance towards longer
wavelengths (i.e., a red, or bathochromic, shift) could substantially increase the amount of light
absorbed by a compound, increasing its reaction rate.  Several studies have measured absorbance
shifts for compounds in LLRs and at the air-ice interface relative to solution (Corrochano et al.
2017, Heger et al. 2005, Kahan and Donaldson 2010, Kania et al. 2014, Krausko et al. 2015,
Malongwe et al. 2016, Matykiewiczová et al. 2007).  Although the absorbance of some
compounds was the same as in aqueous solution, other chemicals shifted by up to 15 nm either to
the red or blue (i.e., a hypsochromic shift).  Unfortunately, measuring a compound's absorbance
at the air-ice interface can be challenging.  Accurate absorbance measurements typically require
relatively high concentrations, which can lead to aggregation on the ice surface, potentially
influencing the absorption characteristics.  To avoid this problem, we recently used molecular
modeling to estimate the absorbance shift for guaiacol at the air-ice interface (Bononi et al. 2020,
Hullar et al. 2020).  While we did find a slight bathochromic shift (5 nm), this shift explained
less than 10% of the enhanced reaction rates found at the interface.
Finally, an increased quantum yield at the air-ice interface could explain a faster reaction rate,
due to a greater fraction of absorbed photons resulting in loss of the chemical.  Some studies
suggest LLRs and solution represent similar reaction environments (Chu and Anastasio 2003,
Chu and Anastasio 2005, Chu and Anastasio 2007, Ram and Anastasio 2009), while others have
found higher quantum yields at the air-ice interface (Hullar et al. 2020, 2018, Zhu et al. 2010).
Our recent work with guaiacol (Hullar et al. 2020) found that changes in the quantum yield were
the dominant contributor to reaction rate differences between aqueous solution, LLRs, and
QLLs, with values up to 40-fold higher at the air-ice interface compared to solution.
Taken together, previous studies show the importance of determining various factors to
understand the reasons for enhanced chemical reaction rates in snow and ice.  In particular, our
recent results (Hullar et al. 2020) indicate that the direct photodecay of guaiacol is different in
aqueous solution, LLRs, and QLLs, and demonstrated how molecular modeling can be used to
assess the relative contributions of changes in light absorbance and quantum yield.  Here, we
extend those results to three additional organic compounds: 1,2-, 1,3-, and 1,4-
dimethoxybenzene (abbreviated 1,2-DMOB, 1,3-DMOB, and 1,4-DMOB, respectively).  1,2-
and 1,3-DMOB have been reported to photodegrade slowly, with 1,4-DMOB loss being
somewhat faster (Amaine et al. 1993).  We measure the direct photochemical reaction rate
constants of these compounds in aqueous solution, LLRs, and QLLs, normalizing each to the
measured photon flux for a given sample type.  To assess the contribution of absorbance shifts,





we model DMOB absorbance in aqueous solution and on an ice surface. As with guaiacol, the
DMOBs are all doubly-substituted aromatic rings; however, the hydroxyl group of guaiacol is
replaced by a methoxy group, eliminating the possible unwanted reaction with triplet excited
states ($^3C^*$). Further, the three isomers provide the opportunity to examine how structural
differences can influence light absorption, quantum yields, and ultimately photochemical
reactivity. To more broadly examine the importance of changes in light absorption on
photodecay, we also quantify how changes in the absorbance peak location, maximum
absorbance, and absorption peak shape affect photochemical reaction rate constants and
lifetimes.

**2 Methods**
**2.1 Materials**
1,2-, 1,3-, and 1,4-DMOB (99, >98, and 99% purity, respectively) were from Aldrich.
Acetonitrile (HPLC grade) was from Acros. 2-nitrobenzaldehyde (2NB, 98%) was from Sigma-
Aldrich. High purity water ("MQ") was from house-produced reverse osmosis water run through
a Barnstead International DO813 activated carbon cartridge and a Millipore Milli-Q Advantage
A10 system ($\geq$ 18.2 M$\Omega$ cm).
**2.2 Sample preparation**
We placed samples in 10-ml glass beakers (Pyrex) and covered them with nylon film
(McMaster-Carr, approximately 25 μm thick, secured in place with an o-ring) to reduce
evaporation and contamination while allowing sample illumination. As discussed in our
previous work with guaiacol (Hullar et al. 2020), we prepared samples using one of five different
methods: 1) in an aqueous solution, where we dissolved the test compound in MQ water to give a
final concentration of 1.0 μM, then we placed 10 ml of solution in a beaker and covered. 2)
Freezer frozen solution, prepared identically to aqueous solution, then placed in a laboratory
freezer (-20 °C) for at least 3 hours. 3) Liquid nitrogen frozen solution, which we prepared
identically to aqueous solution, then placed it in a pan filled to a depth of 2 cm with liquid
nitrogen; sample freezing took approximately 90 seconds. 4) Vapor deposition of gas-phase test
compound to the surface of ice. Following an approach previously described (Hullar et al. 2020,
Hullar et al. 2018), we placed 10 ml of MQ water in a beaker, covered it with film, and froze it
in a laboratory freezer. We removed and uncovered the frozen samples, and directed a nitrogen
stream containing gas-phase dimethoxybenzene at the ice surface for 15 or 30 s. We then
recovered the samples and placed them back in a laboratory freezer. 5) Vapor deposited to
nature-identical snow, which was produced as described in our previous work (Hullar et al.
2020). To deposit dimethoxybenzene onto the snow, we passed nitrogen from a tank in the cold
room first through 500 ml of laboratory-made snow (to condition the nitrogen stream with water
vapor), then through a glass container holding 0.4 g of DMOB, and then through a 500- or 1000-
ml HDPE bottle holding the snow to be illuminated. 1,2-DMOB is a liquid at room temperature
but a solid at -20 °C, while 1,3-DMOB is a liquid at both temperatures and 1,4-DMOB a solid.
We then gently mixed the treated snow and transferred it to beakers, tamped it down 10 mm
below the top edge of the beaker, and covered it with nylon film.

**2.3 Sample illumination, actinometry, and chemical analysis**



Sample illumination followed the method described previously (Hullar et al. 2020). We set
sample beakers upright in a drilled aluminum block set within a temperature-controlled chamber;
dark samples were completely covered with aluminum foil and placed in the aluminum block
next to the illuminated samples. The samples were held at 5 °C (for aqueous) or -10 °C (for ice
and snow). The light source was a 1000 W Xenon arc lamp filtered with an AM1.5 airmass filter
(Sciencetech) and a 295-nm longpass filter (Andover Corporation) to approximate polar sunlight
and a 400-nm shortpass filter (Andover Corporation) to reduce sample heating.
After illumination, we melted the frozen samples and measured DMOB concentration using a
Shimadzu HPLC (Hullar et al. 2018) with an eluent of 60:40 acetonitrile:MQ water, a flow rate
of 0.70 ml min$^{-1}$, and detection wavelengths of 274, 273, and 287 nm for 1,2-, 1,3-, and 1,4-
DMOB, respectively.
To account for differing photon fluxes across samples types and experiment days, we used 2-
nitrobenzaldehyde (2NB) as a chemical actinometer (Galbavy et al. 2010, Hullar et al. 2020,
Hullar et al. 2018). Except for snow samples, we prepared 10 µM 2NB actinometry samples on
each experiment day using the same sample preparation and experiment treatment as the test
compound illuminations, although the illumination times were shorter. For snow samples, daily
$j_{2NB}$ was determined by measuring $j_{2NB}$ in aqueous solution and correcting by a previously
determined ratio ($j_{2NB,snow} / j_{2NB,aq} \pm 1 \sigma = 0.38 \pm 0.015$), as described in earlier work (Hullar et
al. 2020).
We used TUV (Madronich and Flocke 1998) to model spectral actinic fluxes for Summit,
Greenland at noon on the summer solstice (subsequently referred to "Summit conditions"). We
used default settings (option 1) except for wavelength interval = 0.1 nm, latitude = 72.6 degrees,
ground surface elevation = 3,200 m, simulation elevation = 3,200 m, total column ozone = 308
Dobson units, surface albedo = 0.93, and four radiative transfer streams. Using methods
described previously (Hullar et al. 2020), we calculated laboratory photon fluxes at 1 nm
intervals from measured relative light intensities and our $j_{2NB}$ values, then interpolated those
fluxes to 0.1 nm resolution.
**2.4 Determining rate constants and quantum yields for DMOB loss**
We determined DMOB photodegradation rate constants using the same approach used for
guaiacol and PAHs (Hullar et al. 2020, Hullar et al. 2018). We illuminated samples with
simulated polar sunlight, periodically removing a beaker (and corresponding dark beaker) for
analysis. To determine the photodegradation rate constant, we first calculated the natural
logarithm of the ratio of the DMOB concentration at time $t$ to the initial DMOB concentration,
then adjusted these ratios by a correction factor (Supplemental Table S1) to account for
differences in photon flux at each sample position (Hullar et al. 2020, Hullar et al. 2018). The
linear regression slope of illuminated samples gives the pseudo-first-order rate constant for loss
during illumination, $j_{DMOB}$; for dark controls, we get the rate constant for dark loss, $k'_{DMOB,dark}$.
To calculate the net loss attributable to photodegradation, we subtracted $k'_{DMOB,dark}$ from $j_{DMOB}$ to
give the dark-corrected experimental photodegradation rate constant $j_{DMOB,exp}$. We then
normalized this value for the experimental photon flux by dividing $j_{DMOB,exp}$ by the daily
measured $j_{2NB}$ value to give the photon flux-normalized photodegradation rate constant, $j*_{DMOB}$.
Full equations are given in our previous work [14,18].





We calculated quantum yields for each DMOB using methods described previously (Hullar et al.
2020). In short, the quantum yield was determined for each DMOB by dividing the dark-
corrected experimental photodegradation rate constant ($j_{DMOB,exp}$) by the measured aqueous
molar absorptivity ($\varepsilon_{DMOB,\lambda}$) and the calculated photon flux in our experimental system. We
determined aqueous solution molar absorptivities for each DMOB ($\varepsilon_{DMOB,\lambda}$, Supplemental Table
S2) by measuring absorbance spectra in five solutions (10-1000 µM) at 25 °C using a UV-
2501PC spectrophotometer (Shimadzu) in 1.0 cm cuvettes against a MQ reference cell. The
calculated quantum yields are an average value over the ranges of 250-317, 250-315, and 250-
341 nm for 1,2-, 1,3-, and 1,4-DMOB, respectively. We chose the low end of this range because
it represents a rough natural minimum of light absorbance for the three compounds and does not
have any photon flux for either Summit conditions or in our experimental system; the upper
cutoff varies for each compound and is the wavelength above which the molar absorptivity is
less than $5 \times 10^{-5}$ M$^{-1}$ cm$^{-1}$. Based on light absorption spectra for the three compounds
(discussed in the results section), the wavelengths driving photodegradation in our experiments
are 270-300, 270-310, and 280-310 nm for 1,2-, 1,3-, and 1,4-DMOB, respectively. We
estimated molar absorptivities at the air-ice interface by applying the results of the computational
modeling to the measured aqueous molar absorptivities, as described in the results section below.
**2.5 Computational methods**
To investigate possible shifts in light absorbance at the air-ice interface for the three
dimethoxybenzene isomers, we used a multimodel approach that combines classical and first-
principles molecular dynamics (FPMD) simulations, excited state calculations using time-
dependent density functional theory (TDDFT), and machine learning (ML) (Bononi et al. 2020,
Tibshirani 2011).
As in our recent work on phenol and guaiacol, models of DMOB in aqueous solutions and at the
ice surface were equilibrated in classical MD simulations using the OPLS forcefield and the
TIP4P/ice water model (Abascal et al. 2005). To model the air-ice interface we utilized an ice
slab model, which captures a well-equilibrated surface structure and reproduces recent
measurements for QLLs (Kling et al. 2018, Sanchez et al. 2017). We then performed FPMD
simulations of the DMOB isomers in solution at 27 °C and at the air–ice interface at -10 °C.
Density functional theory (DFT)-based FPMD simulations were run using the Perdew-Burke-
Ernzerhof exchange and correlation functional with D3 van der Waals corrections, a double-Z
valence polarizable basis set for valence states and norm-conserving pseudopotentials for the
core states and the nuclei, as implemented in the CP2K code (Grimme et al. 2010, Hartwigsen et
al. 1998, Perdew et al. 1996, VandeVondele et al. 2005). For each 50 ps FPMD simulation
trajectory, we extracted approximately 200 statistically independent frames, replaced the
explicit water molecules with the self-consistent continuum solvation scheme (Timrov et al.
2015), and finally computed absorption spectra for each frame using TDDFT as implemented in
the Quantum-Espresso package (Casida et al. 2009, Giannozzi et al. 2017, Rocca et al. 2008).
To account for the configurational sampling at finite temperature in the specific solvation
environment, we computed the final spectra by ensemble averaging the 200 single frame
calculations for each isomer (Ge et al. 2015, Timrov et al. 2016).
As a refinement to our former approach, we developed a universal ML model to predict the
absorption spectra for all three DMOB isomers. To accomplish the transferability, we adopted a
more sophisticated atomic descriptor - the Bispectrum Component (BC) (Bartok et al. 2013,





Thompson et al. 2015). BC describes each molecule's atomic environment by projecting the
weighted atomic densities to four-dimensional hyperspherical harmonics, and it has been
previously applied to ML interatomic potential development and material property predictions
(Cusentino et al. 2020, Legrain et al. 2017). By using BC with the least absolute shrinkage and
selection operator (LASSO) regression model (Tibshirani 2011), we attain a more precise
estimate of the low-energy, long-wavelength tails of the spectra, which are important for
calculating rates of photon absorption since the photon flux is increasing in this region. To
assess the relative contributions of the phenyl ring and methoxy groups to the light absorbance of
each DMOB isomer, we decomposed the predicted peak wavelength from over ~5000 frames of
each FPMD trajectory, with $\lambda_0 \approx 586$ nm fitted as the intercept of the ML model. Further details
about the simulation procedures and parameters to compute BC and analysis of excitation energy
predictions from the LASSO model are available Supplemental Section S1.



**3. Results**
**3.1. Illumination experiments**
As described in sections 2.2 and 2.3, we prepared samples using one of several methods designed
to place the DMOB isomer into aqueous solution, LLRs, or at the air-ice interface. Then, we
illuminated the samples, periodically removing them for analysis. Supplemental Figures S1
through S12 show the results for each illumination experiment, with each data point representing
one sample beaker. Generally, dark controls show slight loss of DMOB, probably attributable to
volatilization; illuminated samples often show considerably greater loss, presumably due to
photodegradation, but the extent of loss depends on DMOB isomer and sample preparation
method. Within an experiment, sample-to-sample variability was generally less for aqueous,
LN2, and VD-to-snow sample types (e.g., Supplemental Figures S1c, S3b, and S5d), while
frozen solution and VD-to-ice experiments showed considerable experimental noise
(Supplemental Figures S2b and S4c). Previous work (Hullar and Anastasio 2016) suggests more
homogeneous solute distribution in LN2 samples compared to frozen solution samples, which
may explain the reduced experimental variability in LN2 samples. This reduced variability
might also be due to the fact that freezing in LN2 is fast (less than 90 seconds), which reduces
the time available for reactions as solutes are concentrated during freezing, while using a freezer
takes much longer, typically several hours. As discussed for our previous experiments using
nature-identical snow (Hullar et al. 2020), the specific surface area (SSA) for our VD-to-snow
samples is much higher than in VD-to-ice samples, which should reduce aggregation and give
more robust experimental results.
**3.2. DMOB photodegradation for each sample preparation method**
Figure 1 summarizes the experimental results for each of the three DMOBs in aqueous solution
and the various frozen sample preparations. As described above, we divided each dark-
corrected, measured rate constant for DMOB loss by the corresponding measured $j_{2NB}$ value to
compensate for the different photon fluxes in each experiment, then calculated the average





photon-flux-normalized rate constant ($j^*_{DMOB}$) for each sample treatment; error bars in Figure 1
are the 95% confidence interval of mean $j^*_{DMOB}$ values.

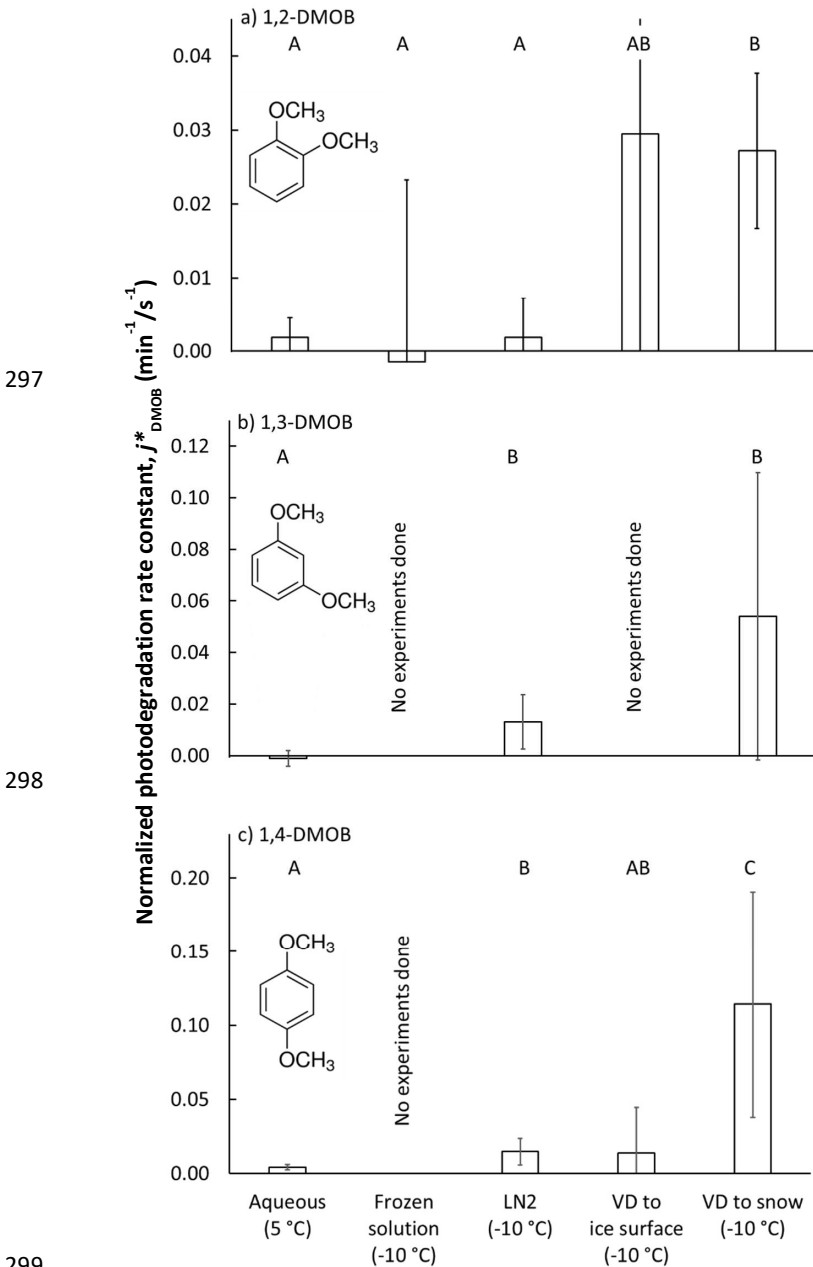









**Figure 1.** Photon-flux-normalized photodegradation rate constants for each dimethoxybenzene isomer ($j*_{DMOB}$) under five experimental conditions: i) aqueous solution, ii) solution frozen in a laboratory freezer ("Frozen solution"), iii) solution frozen in liquid nitrogen ("LN2"), iv) vapor-deposited DMOB to a water ice surface ("VD to ice surface"), and v) vapor-deposited DMOB to nature-identical snow ("VD to snow"). We illuminated samples at 5 °C (aqueous samples) or -10 °C (all others). Bars indicate the mean value for each sample preparation method ($n = 3 - 8$), with error bars as 95% upper and lower confidence limits of the mean (UCL and LCL). For each isomer, sample types having statistically indistinguishable average rate constants as determined by a Tukey-Kramer test ($P < 0.05$) are labeled with the same capital letter ("A", "B", or "C"); sample types with different letters have statistically different means.

As shown in Figure 1a, the 1,2-DMOB photodegradation rate in aqueous solution is slow and the normalized rate constant is statistically indistinguishable from zero. For frozen solution experiments, the average rate constant was negative, and the data was quite noisy. Samples frozen with liquid nitrogen ("LN2") should, like freezer-frozen samples, place solutes primarily in internal LLRs. However, the variability in 1,2-DMOB LN2 experiments is considerably less than for freezer frozen experiments, and the rate constant is roughly equivalent to that determined for aqueous solution. For both frozen solution and LN2 treatments, the rate constants are indistinguishable from zero. The two treatment methods which put 1,2-DMOB at the air-ice interface, VD to ice and VD to snow, both show normalized rate constants approximately 15 times faster than in aqueous solution or in LLRs. However, while experimental results for the VD-to-ice treatment are highly variable (with an average rate constant indistinguishable from zero), VD-to-snow experiments are more reproducible and give a normalized rate constant statistically greater than zero, showing the advantage of using nature-identical snow to study photodegradation at the air-ice interface.

1,3-DMOB results are summarized in Figure 1b. Because the frozen solution and VD-to-ice experiments were very noisy for 1,2-DMOB, we did not run experiments with these sample treatments for 1,3-DMOB. For aqueous solution, the 1,3-DMOB average rate constant is slightly negative and indistinguishable from zero. In LLRs (LN2 sample treatment), 1,3-DMOB photodegrades at a moderate rate, statistically greater than zero. Finally, at the air-ice interface (VD to snow samples), the photodegradation rate constant is approximately four times faster than in LLRs, although statistically indistinguishable from zero because of very high variability.

For 1,4-DMOB in aqueous solution (Figure 1c), the average photodegradation rate is slow, but statistically greater than zero. As with 1,3-DMOB, we did not run experiments in frozen solution for 1,4-DMOB; however, LN2 experiments, which should also place solutes primarily in LLRs, showed photodecay rates both statistically greater than zero and approximately 3-fold faster than in aqueous solution. Measured VD-to-ice rates were variable, and although the average normalized rate constant was similar to LN2, it was not statistically different than zero. As with 1,2-DMOB, the average 1,4-DMOB photodegradation rate constant at the air-ice interface (VD-to-snow experiments) is considerably faster than in either aqueous or LLR compartments, with a 26-fold enhancement relative to aqueous solution, and is statistically greater than zero.

To determine if the various sample treatment rate constants are statistically different from each other, we used the Tukey-Kramer test for multiple comparisons ($P < 0.05$) to generate statistical groupings, identified by the letters A, B, and C on Figures 1a-c. For 1,2-DMOB, mean rate constants ($j^*_{DMOB}$) for aqueous, frozen solution, and LN2 samples were indistinguishable from each other. However, VD to snow gave a rate constant significantly greater than these three



sample types.  Because of its high experimental variability, VD to ice could not be distinguished
from any of the other sample treatments.  For 1,3-DMOB, aqueous samples were statistically
different than both LN2 and VD to snow samples.  However, LN2 and VD-to-snow samples
could not be distinguished from each other.  For 1,4-DMOB, VD-to-snow samples were
statistically higher than every other sample type.  VD-to-ice samples were indistinguishable from
both aqueous and LN2 samples, although LN2 samples were statistically different, and higher,
than the aqueous samples.
Table 1 presents the rate constant enhancements for each frozen sample type relative to aqueous
solution; Supplemental Table S3 provides details for the various measured and computed
experimental parameters.  For 1,2-DMOB, photodegradation proceeds at approximately the same
rate in LLRs and aqueous solution, but roughly 15-fold ($\pm$ 9.5, 1$\sigma$) faster at the air-ice interface.
Because the average aqueous rate constant for 1,3-DMOB was negative, we calculated an upper-
bound estimate (average + 95% UCL of the mean) for the rate constant and determined
enhancements relative to that value.  Compared to the aqueous rate constant, we estimate the 1,3-
DMOB rate constant in LLRs to be at least 6.9-fold faster, and at least 29-fold faster at the air-ice
interface.  Finally, for 1,4-DMOB, enhancement in LLRs is 3.4-fold ($\pm$ 2.4), and 26-fold ($\pm$ 27)
at the air-ice interface.  As noted in previous work (Hullar et al. 2020), vapor depositing a test
compound to nature-identical snow rather than to an ice pellet surface is more representative of
environmental conditions and gives more reliable experimental results, probably due to the much
greater SSA of the snow.  Although the experimental data show considerable variability, our
results suggest DMOB photodegradation rate constants are somewhat faster in LLRs than in
corresponding aqueous solution, and considerably faster at the air-ice interface; this finding is
similar to previously reported results with guaiacol (Hullar et al. 2020) and several other organic
compounds (2007, Kahan et al. 2010, Kahan et al. 2010).  Taken together with earlier work, our
results here suggest that at least for some compounds, aqueous solution, LLRs, and the air-ice-
interface can be different environments for photochemical reactivity.





**Table 1.** Summary statistics for each experimental preparation method[a]

| | | Experimental results | | | Summit conditions estimates | | |
| --- | --- | --- | --- | --- | --- | --- | --- |
| | $n^b$ | $j*_{DMOB}{}^c$ | Enhancement[d] | Quantum Yield $(\Phi_{DMOB})^e$ | Compartment[g] | $j_{TUV,\,DMOB}{}^h$ | Lifetime[i] |
| | | (min$^{-1}$/s$^{-1}$) | $(j*_{DMOB,\,i}/j*_{DMOB,\,aq})$ | (mlc photon$^{-1}$) | | (s$^{-1}$) | (d) |
| **1,2-DMOB** | | | | | | | |
| Aqueous solution | 3 | $0.0019 \pm 0.0011$ | 1 | $0.015 \pm 0.0085$ | Aqueous | $(5.0 \pm 2.9) \times 10^{-10}$ | $23000 \pm 13000$ |
| Freezer frozen solution | 3 | $-0.0010 \pm 0.0099$ | $-0.79 \pm -5.3$ | $-0.012 \pm 0.078$ | | | |
| Liquid nitrogen frozen solution | 4 | $0.0019 \pm 0.0030$ | $1 \pm 1.9$ | $0.015 \pm 0.027$ | LLR | $(5.1 \pm 9.0) \times 10^{-10}$ | $23000 \pm 41000$ |
| Vapor-deposited to ice surface | 3 | $0.029 \pm 0.022$ | $16 \pm 15$ | $0.13 \pm 0.10$ | | | |
| Vapor-deposited to snow | 5 | $0.027 \pm 0.0084$ | $15 \pm 9.5$ | $0.12 \pm 0.039$ | QLL | $(6.3 \pm 1.9) \times 10^{-9}$ | $1800 \pm 570$ |
| **1,3-DMOB** | | | | | | | |
| Aqueous solution | 6 | $-0.0011 \pm 0.0029$ | $1^f$ | $< 0.016$ | Aqueous | $< 2.6 \times 10^{-10j}$ | $> 45000^j$ |
| Freezer frozen solution | 0 | | | | | | |
| Liquid nitrogen frozen solution | 3 | $0.013 \pm 0.0042$ | $> 6.9$ | $0.11 \pm 0.035$ | LLR | $(1.8 \pm 1.4) \times 10^{-9}$ | $6400 \pm 5100$ |
| Vapor-deposited to ice surface | 0 | | | | | | |
| Vapor-deposited to snow | 5 | $0.054 \pm 0.045$ | $> 29$ | $0.085 \pm 0.070$ | QLL | $(2.4 \pm 2.5) \times 10^{-7}$ | $48 \pm 50$ |
| **1,4-DMOB** | | | | | | | |
| Aqueous solution | 3 | $0.0043 \pm 0.00073$ | 1 | $0.0020 \pm 0.00042$ | Aqueous | $(1.6 \pm 0.34) \times 10^{-7}$ | $70 \pm 14$ |
| Freezer frozen solution | 0 | | | | | | |
| Liquid nitrogen frozen solution | 3 | $0.015 \pm 0.0036$ | $3.4 \pm 2.4$ | $0.0075 \pm 0.0018$ | LLR | $(6.0 \pm 1.5) \times 10^{-7}$ | $19 \pm 4.7$ |
| Vapor-deposited to ice surface | 5 | $0.014 \pm 0.025$ | $3.2 \pm 6.1$ | $0.0064 \pm 0.011$ | | | |
| Vapor-deposited to snow | 8 | $0.11 \pm 0.091$ | $26 \pm 27$ | $0.052 \pm 0.042$ | QLL | $(4.4 \pm 3.5) \times 10^{-6}$ | $2.7 \pm 2.1$ |

[a] Samples were held at 5 °C (aqueous samples) or -10 °C (all other preparations).
[b] Number of experiments.

[c] Listed $j*_{DMOB}$ values (photon-flux normalized photodegradation rate constants) are means ± 1 standard deviation.



[d] Enhancement factors are the ratio of the mean $j^*_{DMOB}$ value for each preparation method to the mean aqueous $j^*_{DMOB}$ value for that light condition, ± the propagated standard deviation.

[e] Quantum yields are calculated individually for each experiment using the measured $j_{DMOB,exp}$ and $j_{2NB}$. Uncertainties for quantum yields are ± 1 standard deviation.

[f] To calculate enhancement factors, we first estimated the upper bound $j^*_{DMOB}$ value for aqueous solution as the mean + the 95% UCL, 0.00190 $min^{-1}/s^{-1}$. Then, we calculated enhancement factors relative to this value.

[g] For purposes of calculating $j^*_{TUV,DMOB}$ and photochemical lifetimes, quantum yields in aqueous, LLR, and QLL compartments were assumed to be represented by aqueous solution, liquid nitrogen frozen solution, and vapor-deposited to snow sample types respectively.

[h] Listed $j^*_{TUV,DMOB}$ values (calculated photodegradation rate constants for Summit, Greenland) are means ± 1 propagated standard deviation.

[i] Photochemical lifetimes are 1 / $j^*_{TUV,DMOB}$ values ± 1 propagated standard deviation.

[j] $j^*_{TUV,DMOB}$ and photochemical lifetime calculated from upper-bound estimate of 1,3-DMOB quantum yield.




### 3.3 DMOB light absorbance, quantum yields, and environmental lifetimes in solution and at the air-ice interface

Figure 2 presents the wavelength-dependent molar absorptivities for 1,2-, 1,3-, and 1,4-DMOB,
as well as guaiacol (which was studied in our previous work (Hullar et al. 2020)).  1,2- and 1,3-
DMOB in solution have nearly identical absorbance curves, with maximum absorbance at 274
and 273 nm, respectively.  While guaiacol absorbs less strongly, its curve shape and peak
location are similar to 1,2- and 1,3-DMOB.  In contrast, 1,4-DMOB absorbs at longer
wavelengths, with a peak absorbance at 287 nm.  For comparison, the two black lines in Figure 2
show the photon flux of our experimental system (dashed line) and the modeled actinic flux for
Summit conditions (solid line); a more detailed graph is shown in Supplemental Figure S13.
While the actinic flux at Summit starts at approximately 297 nm and increases quickly with
increasing wavelength, the experimental flux begins earlier (roughly 280 nm) and increases more
gradually.  1,2- and 1,3-DMOB in solution absorb small amounts of light under our illumination
conditions and virtually none in the Arctic environment.  In contrast, the 1,4-DMOB absorbance
curve has substantial overlap with both photon flux curves and therefore absorbs light under both
experimental and natural conditions.

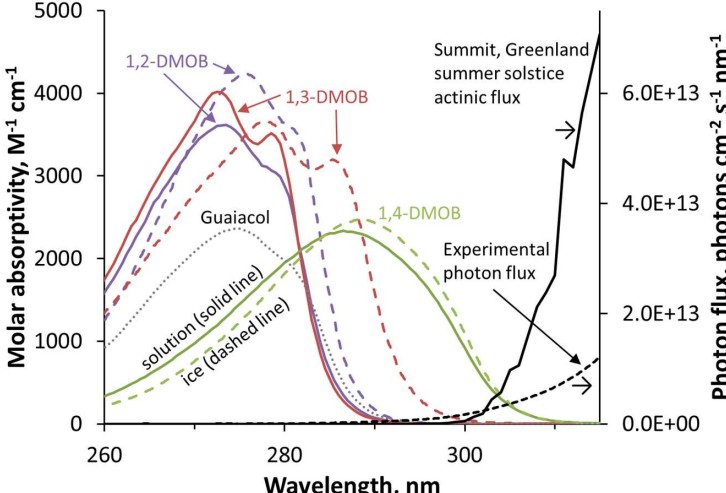


**Figure 2.**  Light absorption spectra for the dimethoxybenzene (DMOB) isomers and guaiacol, along with
photon fluxes in our experiments and for Arctic summer conditions.  Solid colored lines are the measured
molar absorptivities for each DMOB isomer, while colored dashed lines are predicted absorbance spectra
at the air-ice interface, estimated using the results of our molecular modeling.  The solution guaiacol
spectrum (dotted grey line) is provided for comparison to previous work (Hullar et al. 2020).  Black lines
(right axis) represent the modeled actinic flux for Summit conditions (solid line), and the photon flux
measured in our laboratory illumination system (dashed line).





While we can measure light absorption by the DMOB isomers in solution, we also want to
understand their absorption at the air-ice interface.  To estimate this, we use molecular modeling
combined with machine learning for each compound in aqueous solution and at the air-ice
interface; these modeled curves are shown in Figure 3.  As shown in Supplemental Figure S14,
modeled absorbance bands for aqueous DMOBs peak at longer wavelengths (7 to 21 nm)
compared to measurements, equal to or greater than the 7 nm difference we observed for
guaiacol (Bononi et al. 2020, Hullar et al. 2020).  These differences are caused by systematic
underestimation in our simulations, which is a known limitation of TDDFT calculations; the
peak wavelength offset relative to measured spectra tends to increase with larger molecules
(Leang et al. 2012, Miura et al. 2007), consistent with the greater difference here for the DMOB
isomers compared to our previous work with guaiacol.  These differences can be corrected by
applying the same shifts to both solution and ice spectra (Ge et al. 2015).

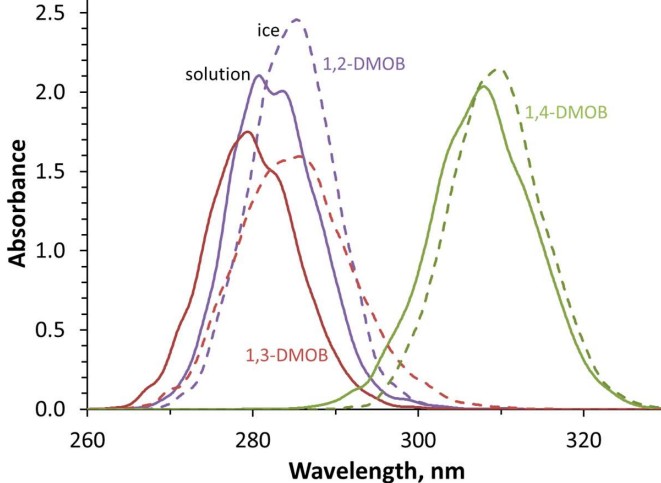

**Figure 3.** Modeled absorbance spectra in aqueous solution (solid lines) and at the air-ice interface (dashed
lines) for each DMOB isomer.  Absolute absorbance values are arbitrary, but accurately reflect the
relative absorbance differences between isomers and conditions.  Temperatures were 27 °C for aqueous
solution and -10 °C at the air-ice interface.

While the modeling does not accurately reproduce the absolute wavelengths of absorbance, it
provides useful insights into the differences between absorbance in aqueous solution and at the
air-ice interface. To predict the absorbance spectrum for each DMOB at the air-ice interface, we
first examine how the model predicts the absorbance changes from aqueous to air-ice interface,
then apply this change to the measured aqueous spectrum.  We quantify how absorbance for a
given DMOB changes from aqueous to ice using three characteristics – peak location, maximum
peak height, and the full width of the peak at half maximum height (FWHM).  In terms of the
first characteristic, all three compounds showed a bathochromic (red) shift at the air-ice interface
relative to aqueous solution; shifts were 2.4, 5.2, and 1.6 nm for 1,2-, 1,3-, and 1,4-DMOB,
respectively.  These results are consistent with a 5-nm red shift modeled for guaiacol (Hullar et
al. 2020) and previous observations of anisole showing a 4-nm red shift at the air-ice interface


(Malongwe et al. 2016), but less than the 10-15 nm red shifts observed for three aniline
derivatives (Corrochano et al. 2017).  For peak height, modeled absorbance peaks of 1,2- and
1,4-DMOB were higher at the air-ice interface compared to in solution, by 17 and 6%
respectively, while the 1,3-DMOB peak height on ice was 91% of the modeled aqueous value.
In terms of peak width, 1,2- and 1,4-DMOB had narrower peaks on ice, 94 and 92% of the
aqueous FWHM, respectively, while 1,3-DMOB had a 27% wider peak on ice.  Using this
information, we applied the modeled peak shifts, peak height changes, and FWHM differences to
the measured aqueous absorbance spectrum for each compound.  This results in predicted
absorbance spectra at the air-ice interface for each DMOB isomer, which are shown in Figure 2
(dashed colored lines) and in Supplemental Figure S14.
We also used the molecular model results to assess the relative contributions of the phenyl ring
and methoxy groups to the light absorbance of each DMOB.  As indicated in Supplemental
Figure S15, small geometrical changes in the phenyl ring are primarily responsible for the shifts
in the absorption spectra for all three DMOB isomers, while the methoxy groups make a minor
contribution.  Changes in the geometry of the phenyl ring are responsible for 95 - 98% of the
light absorbance shifts in aqueous solution and 96 - 98% at the air-ice interface.  These findings
are consistent with our previous work on guaiacol (Bononi et al. 2020, Hullar et al. 2020).
Overall, these results suggest that differences in the atomic environments around the aromatic
ring modify its geometry and determine their vertical excitation and are the primary factor
controlling light absorption changes between aqueous solution and the air–ice interface.
As seen in Figure 2, the predicted spectrum for each isomer at the air-ice interface (dashed
colored line) is noticeably different than the measured aqueous spectrum (solid colored line),
with bathochromic peak shifts and changes in absorbance spectrum shape.  To assess the impact
of these changes on light absorbance, for each isomer we multiplied the aqueous and air-ice
interface wavelength-specific molar absorptivities by the experimental or Summit photon fluxes
to determine the rate constant for light absorbance at each wavelength (Supplemental Figure
S16).  We then summed the wavelength-specific values to obtain the overall rate constant for
light absorbance in aqueous solution and at the air-ice interface, for laboratory and Summit light
conditions (Supplemental Table S4).  Because all three isomers show bathochromic absorbance
shifts at the air-ice interface relative to aqueous solution, the overall rate constants for light
absorption are generally higher at the air-ice interface.  1,3-DMOB, which has the largest
absorbance spectrum bathochromic shift (5.2 nm), shows the largest change in overall light
absorption, with a 5.3-fold increase relative to aqueous solution for experimental light
conditions; for Summit actinic flux, the rate constant of light absorption increases by a factor of
170 from solution to air-ice interface. Conversely, the light absorption peak for 1,4-DMOB shifts
only slightly from solution to ice and has a greater overlap with the photon flux curves in
solution, so the rate of light absorption increases only slightly (10% or less) from solution to the
air-ice interface.  These results show that the amount of light absorbed can be dramatically
affected by absorbance changes and that this effect depends strongly on the position of the
absorbance spectrum relative to photon fluxes and on the magnitude of the absorbance shift on
ice.  Comparing the overall light absorbed under laboratory versus Summit light conditions, 1,2-
DMOB in either aqueous solution or at the air-ice interface absorbs around 200 times as much
light in our lab system compared to Summit, while for 1,4-DMOB the light absorption is
approximately equal in both systems.  1,3-DMOB presents a more complex picture: in aqueous
solution, the rate constant of light absorption is about 400-fold greater under laboratory



illumination compared to Summit light conditions, but at the air-ice interface, light absorption is
only 12-fold greater in the lab relative to Summit conditions due to the absorbance shift on ice.
For 1,2- and 1,3-DMOB, wavelengths from 275 to 295 and 295 to 315 nm are most
photochemically important for lab and Summit light conditions, respectively; for 1,4-DMOB,
these ranges are 280-315 and 300-320 nm.
Our observed increases in photochemical degradation rates at the air-ice interface can be caused
by increases in light absorbance, quantum yield, or a combination of both. As shown previously
(Hullar et al. 2020), by solving equation (1) for quantum yield we can use the calculated
enhancements in the rate constant of light absorbance from our modeling results to estimate how
quantum yields change from solution to the air-ice interface. Using the measured aqueous and
predicted ice spectra for each compound, we calculated the quantum yields for each isomer
under various conditions (Table 1). Our experimental results suggest LLRs may represent an
environment different from either aqueous solution or QLLs. However, we did not model light
absorbance changes in LLRs, so for the quantum yield calculations we assumed our test
compounds have the same molar absorptivities in LLRs as in aqueous solution.
For 1,2-DMOB, the quantum yield in aqueous solution is 0.015 ($\pm$ 0.0085, 1 $\sigma$). Because the
experimental data is noisy, the calculated quantum yield for frozen solution is statistically
indistinguishable from zero. LN2 samples had a similar quantum yield to aqueous solution,
although again the quantum yield is indistinguishable from zero. In both VD-to-ice and VD-to-
snow samples, where we would expect to find 1,2-DMOB at the air-ice interface, the quantum
yields were approximately 8 times higher than in aqueous solution or LLRs, and were the highest
calculated for any isomer and sample type, e.g., $0.12 \pm 0.039$ in the VD-to-snow samples. For
1,3-DMOB, the negative experimental reaction rate constant in solution (Figure 1b) precludes
calculating a quantum yield; however, using the calculated confidence interval (Table 1) we can
provide an upper-bound estimate of 0.016. In both LN2 (LLR) and VD-to-snow (air-ice
interface) sample types, the 1,3-DMOB quantum yields are similar, and at least 5 times higher
than in aqueous solution. For 1,4-DMOB, the aqueous solution quantum yield of 0.0020 ($\pm$
0.00042) is approximately 8-fold less than that of 1,2-DMOB; the quantum yield in LLRs is
approximately 4 times higher than in aqueous solution. At the air-ice interface, the 1,4-DMOB
quantum yield ($0.052 \pm 0.042$ mlc photon$^{-1}$) had the largest increase in quantum yield relative to
aqueous solution of any of the isomers, approximately 26-fold. These results are in the same
range as previous results showing 40- and 3-fold increases in air-ice interface quantum yields for
guaiacol and nitrate, respectively (Hullar et al. 2020, 2018).
Next, we evaluated the relative contributions of increased light absorbance and larger quantum
yields to the photodegradation rate enhancements at the air-ice interface relative to solution. For
1,2- and 1,4-dimethoxybenzene, the faster photodegradation on ice is primarily due to an
increase in quantum yield. In contrast, for 1,3-DMOB the enhanced photodegradation at the air-
ice interface is roughly equally due to increases in quantum yield and light absorbance. As in
our earlier work with guaiacol (Hullar et al. 2020), light absorbance changes are never the
dominant factor controlling rate constant enhancements. Increased light absorption accounts for
16%, 49% or less, and 4% of the reactivity enhancement on snow relative to aqueous solution for
1,2-, 1,3-, and 1,4-DMOB, respectively. Thus, higher quantum yields account for the bulk of the
enhancement seen at the air-ice interface, accounting for 84%, at least 51%, and 96% of the
observed enhancements, respectively. These results are roughly consistent with our previous
observations for guaiacol, where the quantum yield increased at the air-ice interface by a factor



of 41, accounting for 95% of the overall 77-fold increase in reactivity compared to aqueous
solution (Hullar et al. 2020).

**3.4 Estimated photodegradation rate constants under environmental conditions and**
**sensitivity to absorbance shifts**

To assess the environmental significance of our findings, we calculated dimethoxybenzene
photodegradation rate constants and photochemical lifetimes in each compartment for Summit,
Greenland conditions (Table 1). For these calculations, we used modeled actinic fluxes at
Summit (section 2.3) and our estimated quantum yields (section 3.3); because our computational
modeling did not include LLRs, we used measured aqueous spectra to represent absorbance in
both aqueous and LLR compartments, and our predicted ice spectra (Figure 2) for the air-ice
interface. 1,2-DMOB has slow photodegradation rate constants and very long photochemical
lifetimes (~60 years) in solution and in LLRs. At the air-ice interface, it photodegrades 13 times
faster, but the resulting lifetime is still long – approximately 1800 days of midday, summer
solstice sunlight. Thus even with the rate constant enhancement at the interface, direct
photochemical degradation is still apparently negligible. Similar to 1,2-DMOB, 1,3-DMOB has
long lifetimes and slow degradation rate constants in the aqueous and LLR compartments (Table
1). However, because of its significant bathochromic absorbance shift (5.2 nm), the lifetime of
1,3-DMOB at the air-ice interface decreases to 48 days, suggesting its persistence in the
environment may change significantly depending on where it is found in snow or ice. For 1,4-
DMOB, its absorption is already at longer wavelengths compared to the other two isomers, so it
absorbs more light under environmental conditions and therefore can undergo faster
photodegradation: lifetimes are 70, 19, and 2.7 days in aqueous solution, LLRs, and QLLs,
respectively, under Summit conditions. These results show that the location of an impurity in a
snowpack can strongly influence photochemical degradation rates. For compounds that absorb
sunlight substantially in solution, direct photochemical reactions at the air-ice interface may be
an important transformation process in snowpacks. However, quantifying this effect requires
understanding the portion of a compound at the air-ice interface, which is poorly known.
As discussed above, enhanced reactivity at the air-ice interface is primarily due to increases in
the quantum yield, ranging from at least 5-fold (1,3-DMOB) to 41-fold (guaiacol) (Hullar et al.
2020). However, although we can predict absorbance shifts at the air-ice interface using
molecular modeling techniques, we cannot currently predict quantum yield changes using either
computational or experimental methods. While changes in quantum yields affect
photodegradation rate constants linearly – a doubling of quantum yield will double the rate
constant – absorbance shifts cause nonlinear effects. To evaluate the impact of absorbance shifts
on compound photodegradation, Figure 4a shows the calculated ratios of absorbance-shifted rate
constants to the unshifted rate constant. We estimated environmental ($j_{TUV,DMOB}$) and lab
photodegradation rate constants for each isomer using our calculated aqueous solution quantum
yield, Summit or experimental photon fluxes, and our measured aqueous absorbance including
bathochromic and hypsochromic shifts to simulate absorbance changes in/on ice. For our
experimental photon flux, a 5 nm bathochromic shift (approximately equal to the largest shift
modeled for the three DMOB isomers) changes the photodegradation rate constants by factors of
3.1 and 2.1 for 1,2-DMOB and 1,4-DMOB respectively. A 10 nm shift, similar to that measured
for several aniline derivatives (Corrochano et al. 2017), increases the photodegradation rate
constants by factors of 7.8 and 4.2. Because 1,4-DMOB absorbs more strongly at longer
wavelengths than 1,2-DMOB and therefore has greater initial overlap with the experimental



photon flux (Figure 2), shifts have a smaller impact than for 1,2-DMOB.  For a 5 nm
hypsochromic (blue) shift, 1,2-DMOB and 1,4-DMOB rate constants are reduced by factors of
0.24 and 0.46, respectively; for a 10 nm shift, the factors are 0.040 and 0.19.  As with red shifts,
1,2-DMOB shows greater sensitivity to a blue shift than does 1,4-DMOB.  With Summit actinic
fluxes, we see a similar pattern, but a greater sensitivity to shift (Figure 4a) due to the faster
increase in actinic flux with increasing wavelength for Summit compared to our experimental
photon fluxes (Figure 2).  For 1,2-DMOB, a 10 nm red or blue shift changes the rate constants by
factors of 90 or 0.0078, respectively, under Summit (TUV) photon fluxes.  For 1,4-DMOB, the
same shifts yield changes of 16- or 0.029-fold.  1,3-DMOB has a very similar absorbance
spectrum to 1,2-DMOB, and thus shows similar impacts of a shift in its absorbance spectrum
(Supplemental Figure S17).   Supplemental Figure S18 compares guaiacol photodegradation rate
constant changes estimated using TUV actinic flux and under three different experimental light
source conditions from this and previous work (Hullar et al. 2020), showing how our
experimental illumination system has been improved over time, but still does not fully reproduce
the solar spectrum of Summit conditions.

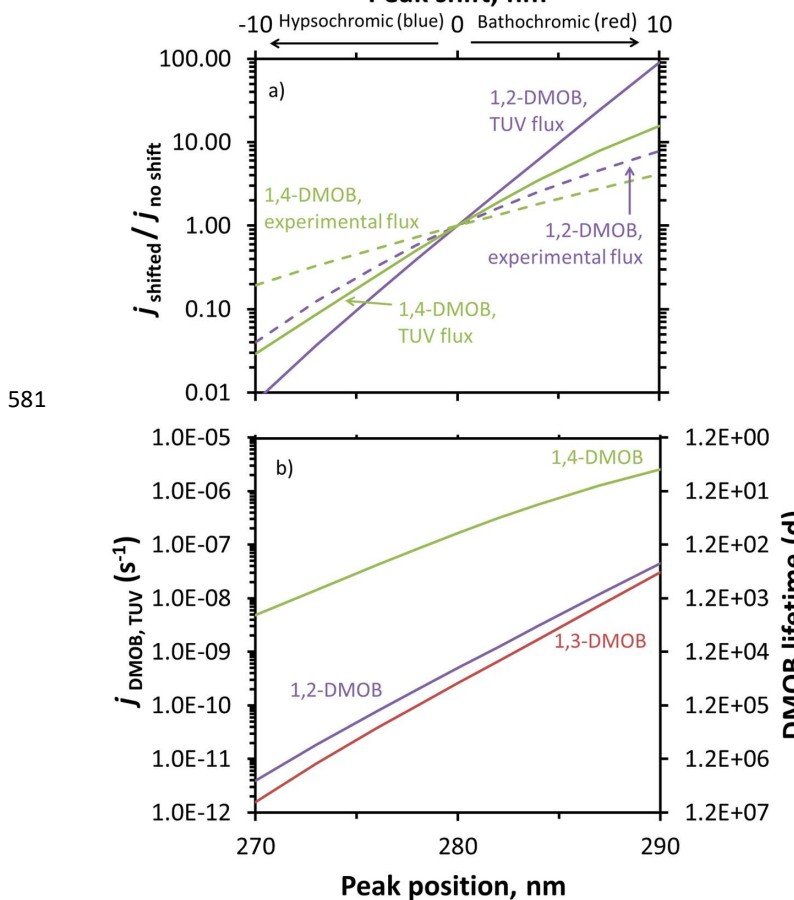



**Figure 4**. Predicted changes to photodegradation rate constants and lifetimes resulting from absorbance shifts for DMOB isomers. Rate constants were determined using calculated aqueous quantum yields, aqueous absorbance spectra shifted either hypsochromically (towards shorter wavelengths) or bathochromically (towards longer wavelengths), and either experimental photon fluxes (dashed lines) or the modeled actinic flux for Summit conditions (solid lines). a) Changes in absorbance-shifted photodegradation rate constants ($j*_{DMOB}$ for experimental conditions, $j*_{DMOB,TUV}$ for TUV-modeled photon fluxes) relative to unshifted values for 1,2- and 1,4-DMOB in aqueous solution. b) Estimated changes in direct photodegradation rate constant ($j_{DMOB,TUV}$) and corresponding lifetime for each DMOB isomer under Summit conditions for various shifts in the light absorbance peak.

While the impact of a red-shift in absorbance can be dramatic, this does not necessarily translate to a short lifetime. For example, a 10-nm red shift for 1,2-DMOB increases the rate constant for photodegradation by a factor of 90 (Figure 4a), but this only reduces the lifetime from 23,000 to 260 days (Figure 4b). 1,3-DMOB, which has essentially the same absorbance spectrum, behaves similarly (Figures 4, 5, and S17). The behavior of 1,4-DMOB is different, however, since it overlaps the most with the solar spectrum (Figure 2): while its rate constant for loss is less sensitive to a shift in absorbance (e.g., increasing by a factor of 16 for a 10-nm red shift), this changes the lifetime from 71 to 4.5 days (Figure 4b), which is short enough to be significant for its environmental fate.

**3.5 Sensitivity analysis of absorbance parameters on photodegradation rate constants for a hypothetical model compound**

To generalize our experimental findings to other chemicals, we calculated photodegradation rate constants and lifetimes for a hypothetical model compound with an assumed Gaussian absorbance spectrum under Summit conditions and with a quantum yield of 1. We first chose a single hypothetical absorbance curve, then evaluated the impact of three variables: peak position, peak width, and peak height. We created a single hypothetical model compound with absorbance represented as a Gaussian curve with its peak at 280 nm, peak height (molar absorptivity) of 3000 $M^{-1}$ $cm^{-1}$, and a standard deviation (controlling peak width) of 7 nm. These parameters were determined by fitting two Gaussian curves to the absorbance spectra for 1,2- and 1,4-DMOB (Supplemental Figure S19); because the right-hand (red side) of each absorbance spectrum determines the amount of light absorbed, we selected curves to fit this portion of the measured absorbance spectra, then applied the average parameters to give the absorption spectrum of the hypothetical compound.

We then evaluated the impacts of shifting the peak position widely, by ± 20 nm, as illustrated in Figure S20. As shown in Figure 5a for Summit sunlight, the impact of a shift depends on the where it occurs, with the rate constant for photodegradation more sensitive at shorter wavelengths. For example, hypsochromically moving a peak from 280 to 260 nm decreases the rate constant by a factor of 320,000, while bathochromically moving the peak from 280 to 300 nm leads to a 920-fold increase in rate constant. This difference is because the red shift moves the absorbance spectrum towards wavelengths where some light is already being absorbed. However, as discussed earlier, large increases in a rate constant do not necessarily translate to significant photochemistry: lifetimes for our hypothetical compound are 2,200,000, 6.7, and 0.0072 days when the peak is centered at 260, 280, and 300 nm, respectively (Figure 5b). This

sensitivity of shift impact to starting peak wavelength is shown even more clearly if we consider
a wider wavelength range, as shown in Figure S21.  Once the compound absorbance begins to
significantly overlap with the actinic flux curve, additional red-shifting does not dramatically
increase the amount of light absorbed, slowing the rate of $j_{max}$ increase.  If we assume our
hypothetical compound experiences a 5 nm shift, the largest shift estimated by our calculations
of the three DMOB isomers, the rate constant would decrease by a factor of 0.0075 for a blue
shift and increase by 9.2-fold for a red shift.


**Figure 5.**  Predicted changes to photodegradation rate constants ($j_{max}$) and lifetimes resulting from
absorbance shifts for a hypothetical model compound.  Rate constants ($j_{max}$) and lifetimes calculated using
an assumed quantum yield of 1, modeled actinic flux for Summit conditions, and an assumed Gaussian
absorbance spectrum (peak molar absorptivity 3000 M$^{-1}$ cm$^{-1}$, standard deviation of 7 nm) with varying
peak positions.  a) Ratio of shifted to unshifted $j_{max}$ for varying hypsochromic (blue) or bathochromic





(red) absorbance shifts.  b) Calculated rate constants ($j_{max}$) and lifetimes at various peak positions.  The
horizontal lines intersect the curves at the baseline peak position of 280 nm.

Next, we examined the impact of peak width, as illustrated in Figure S22.  From our modeling,
the largest peak width change was approximately 2 nm (for 1,3-DMOB).  As seen in
Supplemental Figure S23, narrowing the hypothetical peak from 7 to 5 nm reduces $j_{max}$ and
increases the lifetime by 88-fold, roughly 7 times larger than the lifetime decrease caused by a 5
nm hypsochromic shift of the original 7 nm wide peak.  While broadening the peak to 9 nm does
increase $j_{max}$ and decrease the lifetime, the magnitude of the change is not as significant,
approximately 13-fold.  Similarly to the pattern seen for peak location shifts, changes in peak
width cause greater impacts when the compound's absorbance peak is located at shorter
wavelengths.
Finally, we evaluated the impact of changing the peak height (hyper- and hypochromic shifts).
Figure S24 shows the spectra tested, and Figure S25 the results; for comparison, our largest
modeled peak height change was 17%, for 1,2-DMOB.  Because the area of a Gaussian curve is
proportional to its peak height, doubling the height doubles the area and therefore the light
absorbed would double as well.  However, compared to the impact of peak location and width,
even a peak height doubling exerts a relatively small influence on peak area and therefore light
absorbed.  To evaluate the relative impact of absorbance shifts, broadenings, and peak height
(molar absorptivity) changes on photodegradation, we assumed the largest modeled absorbance
changes between aqueous solution and at the air-ice interface for the three DMOB isomers are
typical for chemicals in the environment.  Based on this assumption and applying these changes
to our hypothetical peak, peak location and width changes at the air-ice interface probably
control overall differences in light absorption, while changes in peak height likely make a minor
contribution.

## 4 Conclusions

Our results, together with previous studies (Hullar et al. 2020, 2007, Kahan et al. 2010, Kahan et
al. 2010), suggest that for some organic compounds, QLLs and LLRs represent different
photochemical reaction environments that are distinct from aqueous solution.  While molecular
modeling and laboratory measurements have both found evidence of absorbance shifts
(Corrochano et al. 2017, Heger et al. 2005, Hullar et al. 2020, Malongwe et al. 2016), our results
indicate that increases in quantum yield are the major reason for enhanced photochemical
reactivity at the air-ice interface.  For compounds absorbing appreciable amounts of sunlight in
aqueous solution, QLL and LLR reactivity increases may cause environmentally significant
changes in direct photoreaction rates and lifetimes, but for chemicals that absorb very little or no
sunlight, these changes do not appear to make direct photochemistry a significant sink.
Our ability to make statistically significant conclusions depended on the choice of the
experimental treatment; samples frozen in liquid nitrogen or vapor deposited to nature-identical
snow provided useful insights into LLR and QLL compartments, respectively.  In contrast,
samples frozen in a laboratory freezer or vapor deposited to a water ice surface gave results that
were noisier and less valuable.  In addition, computational methods allowed us to determine
absorbance spectra at the air-ice interface, where experimental observations would have been
difficult.








**Acknowledgments**

We thank the National Science Foundation for funding (CHE 1806210 and AGS-PRF 1524857).
Calculations were performed using the Extreme Science and Engineering Discovery
Environment (XSEDE),87 which is supported by the National Science Foundation, grant number
ACI-154856



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
