# Peer review of "Enhanced photodegradation of dimethoxybenzene isomers in/on ice compared to in aqueous"

_Atmospheric Chemistry and Physics, 2021_

## Referee Comment (RC2)

Review on:

**Enhanced photodegradation of dimethoxybenzene isomers in/on ice compared to in**
**aqueous solution**

by: Ted Hullar, Theo Tran, Zekun Chen, Fernanda Bononi, Oliver Palmer, Davide Donadio, and Cort Anastasio

The article assesses experimentally the rates of disappearance of three variously substituted dimethoxybenzens in aqueous solutions, frozen in a freezer, liquid nitrogen, and deposited from the vapor phase onto the surfaces of ice and snow. 1,2- and 1,3 – DMOBs do not react in either a liquid or a frozen aqueous solution because they do not absorb applied light; 1,4-DMOB, by contrast, absorbs and reacts. Importantly, the vapor-deposited compounds react in cases of all the isomers. Independently of the experiments, the absorption spectra of the compounds on the ice surface are calculated. The topic is appropriate for atmospheric chemistry and physics and features substantial novelty, as these compounds have not been photolyzed in/on ice to date. The research embodies an extension of the authors' previous papers on different compound (Hullar et al., 2020). I appreciate the fair effort of the authors to contribute towards expanding the knowledge of the photochemical fate of compounds in the frozen state, which is indispensable in describing the fate of the compounds in the cryosphere. The article presents a substantial amount of data. Regrettably, I have certain reservations concerning the quality of the data, correctness of the performed actinometry, and proper mathematical treatment of the kinetics and interpretation. Moreover, I have identified a number of nomenclature issues that are inconsistent with their common use. I suggest that these issues be addressed by the authors prior to the acceptance of the manuscript.

**Conceptual problems:**

1. Equation 1 treats the heterogenous mixture as homogeneous. There is abundant literature to support the heterogeneity of the compounds in the frozen sample (Bartels-Rausch et al., 2014) (Heger et al., 2007). Generally, at least three environments are available: a) surface; b) veins and pools; c) crystals in which the QY (quantum yields) can vary. Regarding the formation of the crystals, the freeze-concentration effect and solubility should be discussed. The photochemistry in the crystals can yield photoproducts different from those in solutions. The authors, however, discuss only the concentrations of the reactants. Could they comment on the photoproducts as well? The heterogeneity of the reacting mixture can result in a non-monoexponential disappearance of the reactants; therefore, the adherence of the experimental data to the supposed monomexponential decay should be shown.

2. The authors should expose that the increased disappearance rates for the VD (vapor deposited) compounds are not due to merely the lower concentration in the irradiated samples. For this purpose, the absolute concentrations of the compounds should be provided.

3. To calculate the QY, the knowledge of the absorbed number of photons is required; I do not assume this is reliably estimable from the current data (for the heterogenous samples and with a large uncertainty of the current disappearance rate constants and absorbance). Thus, I feel that most of the discussion is markedly speculative and should be announce in this way.

4. The absorption spectra were simulated for ice surface. How can the authors ensure that they are correct? Does the applied method reproduce solvatochromic shifts in a solution for some known compounds? Can the simulation be done for an FCS (freeze concentrated solution)? Concerning further issues associated with the simulation of the spectra: I suspect the formation of crystals on the ice surface by vapor deposition (Ondrušková et al., 2018) and frozen (amorphous) solution in the veins by LN freezing. (Ondrušková et al., 2020) Therefore, I think that the simulation should attempt to distinguish the FCS, ice-air interfaces and crystals of the compounds. In other words, even though the simulation could correctly predict interaction of compounds with the ice surface, that does not need to be the main species in the prepared sample. The authors are invited to address this issue.

5. As the employed actinometer, *o*-nitrobenzaldehyde, absorbs and photoreacts up to 366 nm and 404 nm (Leighton et al., 1934), the correction for the DMOBs is very imprecise and possibly inapplicable. I propose that the absorption spectra of the studied compound be plotted together with *o*-nitrobenzaldehyde to show the spectral overlap (or the lack of it). Monochromatic light (filtered low pressure mercury lamp) would be preferable for the actinometry (besides the polar sun experiments).

6. The aggregation of compounds in nature is reported in ice, e.g., (Mulvaney et al., 1988). I agree that both the separated and the aggregated compound can photoreact. It would be worth discussing under which natural circumstance the compounds do and do not aggregate. As a matter of fact, I cannot find in the article a proof that the compounds are not aggregated under the current experimental conditions.

**Suggestions:**

1. The mechanisms, photoproducts, and rate constants for the photochemically induced reactions of DMOBs should be reviewed to allow a mechanistical explanation of the proposed changes.

2. The introduction and results suffer from permanent referencing to previous research of the authors, to such an extent that the present manuscript is not comprehensible without simultaneous reading of the articles published in the past. In this context, I strongly recommend that a full description of the sample preparation be included in particular. For a reader, it is generally inconvenient to have to refer to the authors' previous publications to learn how each sample was prepared.

**Formal and nomenclature problems:**

1. **QLL**: the term is not used properly, e.g., lines 44-45, Table 1, and throughout. QLL (synonyms disordered interface, surface melting…) forms on the surfaces of pure water ice. At a moment of inserting impurities in, the layer is not QLL but a brine layer – that should be recognized in the manuscript. In the manuscript, the QLL is sometimes also confused to specify the ice-air surface. QLL forms in any surface of ice crystals, not just on the ice-air interface but also at grain-boundaries (Wettlaufer et al., 2006); this use of the term is also in (Barret et al., 2011) for example. A quasi brine layer was introduced to describe the brine at subeutectic temperatures (Cho et al., 2002). I strongly recommend that these distinct terms be kept in separate uses to avoid further confusion.

2. I would like to ask the authors to adhere to the correct nomenclature regarding the disappearance of the chemical compounds. Firstly, the "**photochemical reaction rate**

**constant**" (line 122) is not defined. A photochemical reaction starts with the absorption of light and is followed by some relaxations like intersystem crossing, internal conversion, fluorescence, and reactions from the excited state – the individual processes can be characterized by rate constants and quantum yields. The rate constants are typically fast ($k$= 10e7-10e3 s^-1), measured by techniques applying pulsed lasers (Givens et al., 2008; Klán et al., 2009). Such rate constants are, however, not measured here. I feel strong urge to create the nomenclature that avoids any possible confusion between these rate constants and the apparent rates of compounds disappearance. Thus, statements such as that on line 106 "… a faster reaction rate,…" (and many others) should be avoided, as the reaction rate implicitly describes the rate of some reaction step or overall reaction, which is not being determined here.  In reality, the reaction rates can be altered by the freezing; thus, statements like that on line 106 can be interpreted in a other way than that intended by the authors. What the authors are describing (eq. 1) should be, in my view, called the **apparent rate of compound disappearance**. The adjective "apparent" or an applicable synonym would clearly indicate that no rate of chemical reaction or of an elementary step is discussed. The measured apparent rate of compound disappearance depends of the photon flux, distance from the lamp, filters applied, … whereas the rate of the chemical reaction does not.

3. **Molar absorptivity** is not correct term. I suppose the author means "molar absorption coefficient" and should use this term throughout. (https://goldbook.iupac.org/terms/view/M03972)

4. The rate constant for the light absorption is a term without a physical meaning in a discussed context. Light is absorbed instantly (in order of femtoseconds or less).

**Data collection problems**

Importantly, the presented data mostly report insufficient sampling and reproducibility, together with inadequate irradiation times or a too weak light source. For example, Figure S2a shows an increasing concentration of the compound in the dark experiment. How is this explained? I suggest statistical testing if the concentration after the preset time (at the end of the irradiation) significantly differs from that at the beginning. Besides the data in Figures S6f, S7a-c, 10, 11b-c, I do not think the statistical testing would reveal the influence of irradiation (or time in general). In the negative case, the data should not be used in further analysis, and the experimental design should be improved.

Example Figure1c.: The conversion in 1,500 min ranges from ca 0.5 to 2 %. Such a small conversion with this large spread does not allow reliable estimate of the apparent rate constant. If the uncertainty of the slope were calculated, this fact would be apparent.

The data in Figure 1 are presented in such a manner that there does not seem to be any difference in the rate of photodegradation in first three preparation approaches. As a matter of fact, a closer look at Figures S1-12 indicates that under the performed experimental conditions the difference cannot have been detected. This fact must be emphasized in the manuscript. The post hoc Tukey-Kramer test does not reveal the issue. The performed analysis (discussed in the text) corroborates the inconclusive experiments, as most of the outputs are not significant and are highly scattered. This makes the appropriateness of publishing these data questionable.

The negative apparent rates of photochemical degradation should not be reported and should warn the authors about their experimental design.

At the same time, considering my own experience in the preparation of frozen samples, I know that these samples tend to produce heterogenous outputs; therefore, I really value the authors' work. Only, I press on not to conclude from the data the outputs that they do not offer. Still, the difference in the vapor-deposited samples is apparent and worth publishing, on the condition that the absolute concentrations are shown.

**Data treatment problems**

A major question rests in whether the observed rate of the compound disappearance, under steady irradiation, will follow a monoexponential decay. To respond to this issue, there are a number of reasons that indicate why it will not: For example, if the ice creates a non-homogeneous environment where compounds undergo degradation with unequal quantum yields or if the photoproduct would absorb. Importantly, the authors should show the data and analyze their deviations (residuals) from the monoexponential decay (without the logarithmic linearization).

Linearization of the kinetics was required before the availability of computers. Today, there is no basis for the method. Moreover, the exponential fit provides proper weighting of each of the points, whereas linearized one does not. Fitting the exponential decay curve and plotting the data in this way would allow the authors to determine more easily if the fit makes sense and if the sample was exposed to the light long enough.

**Sample preparation:**

Importantly, the absolute concentrations of the compounds in the VD (vapor deposited) samples should be given.

Further, the exact temperature of the vapor deposition may play a role, too: The fluorescence spectroscopy of naphthalene showed that an amorphous layer forms at 77 K, whereas microcrystals emerge at 253 K. (Ondrušková et al., 2018) The monolayer was never detected. I suspect the microcrystals formation under the here reported "laboratory freezer" experimental temperature.

**Particular comments:**

44-46 The sentence is misleading for above mentioned reasons.

96 The cited publication often observes the bathochromic shifts in both frozen and VD samples. Comparison of these differences may be important for the current study.

98-102
Importantly, there is a literature report on 100 nm shift in the absorbance upon freezing. (Heger et al., 2007) As a matter of fact, the aggregation on the ice surfaces in the experiments does always need to be a disadvantage: the same processes are to be expected to occur in the nature during the metamorphosis processes.

124 "normalizing each to the measured photon flux.." I am worried, that the photon flux is measured at the different wavelengths, thus is not much applicable.

168 How was the homogenous irradiation guaranteed? Were the samples rotated? Please, provide some detail, how much variance can be the photon flux and how much variability comes form the sampling and chemical analysis.

204 Why should the dark loss follow first order disappearance? And does it?

312 "normalized rate constant is statistically indistinguishable from zero" This is important. This is saying that the data are too noisy to draw any conclusion from. I think the authors should stress this fact. Still, the possible conclusion is that VD samples react more readily then the aqueous or frozen.

375 The experimental spectra of molar absorption coefficients reported in SI are valuable contributions. There will be even more valuable, if provided with standard errors of means, which should be available to the authors. Please, provide them.

488-490 The absorption properties in FCS is possibly more altered then on the ice surface due to more intensive interactions in that environment.

491 A reference for the QY is missing.

539 …in snow or ice" Could the authors elaborate what changes do they expect on these two environments and why?

608 Gaussian curve fitting. Before the fit, data should be converted to scale with energy. (Antonov, 1997) More than two transitions are apparent – as discussed in the text (611) however only one curve is plotted in the Figure S19.

Antonov, L. (1997). Drawbacks of the present standards for processing absorption spectra recorded linearly as a function of wavelength. *Trac-Trend. Anal. Chem., 16*(9), 536-543.

Barret, M., Domine, F., Houdier, S., Gallet, J. C., Weibring, P., Walega, J., Fried, A., & Richter, D. (2011). Formaldehyde in the Alaskan Arctic snowpack: Partitioning and physical processes involved in air-snow exchanges. *Journal of Geophysical Research-Atmospheres, 116*. doi:D00r03

10.1029/2011jd016038

Bartels-Rausch, T., Jacobi, H. W., Kahan, T. F., Thomas, J. L., Thomson, E. S., Abbatt, J. P. D., Ammann, M., Blackford, J. R., Bluhm, H., Boxe, C., Domine, F., Frey, M. M., Gladich, I., Guzmán, M. I., Heger, D., Huthwelker, T., Klán, P., Kuhs, W. F., Kuo, M. H., Maus, S., Moussa, S. G., McNeill, V. F., Newberg, J. T., Pettersson, J. B. C., Roeselová, M., & Sodeau, J. R. (2014). A review of air–ice chemical and physical interactions (AICI): liquids, quasi-liquids, and solids in snow. *Atmos. Chem. Phys., 14*(3), 1587-1633. doi:10.5194/acp-14-1587-2014

Givens, R. S., Heger, D., Hellrung, B., Kamdzhilov, Y., Mac, M., Conrad, P. G., II, Cope, E., Lee, J. I., Mata-Segreda, J. F., Schowen, R. L., & Wirz, J. (2008). The Photo-Favorskii Reaction of p-Hydroxyphenacyl Compounds Is Initiated by Water-Assisted, Adiabatic Extrusion of a Triplet Biradical *Journal of the American Chemical Society, 130*(11), 3307-3309.

Heger, D., & Klan, P. (2007). Interactions of organic molecules at grain boundaries in ice: A solvatochromic analysis. *Journal of Photochemistry and Photobiology a-Chemistry, 187*(2-3), 275-284. doi:10.1016/j.jphotochem.2006.10.012

Hullar, T., Bononi, F. C., Chen, Z., Magadia, D., Palmer, O., Tran, T., Rocca, D., Andreussi, O., Donadio, D., & Anastasio, C. (2020). Photodecay of guaiacol is faster in ice, and even more rapid on ice, than in aqueous solution. *Environ Sci Process Impacts, 22*(8), 1666-1677. doi:10.1039/d0em00242a

Cho, H., Shepson, P. B., Barrie, L. A., Cowin, J. P., & Zaveri, R. (2002). NMR investigation of the quasi-brine layer in ice/brine mixtures. *Journal of Physical Chemistry B, 106*(43), 11226-11232.

Klán, P., & Wirz, J. (2009). *Photochemistry of Organic Compounds: From Concepts to Practice*: Wiley.

Leighton, P. A., & Lucy, F. A. (1934). The Photoisomerization of the o-Nitrobenzaldehydes I. Photochemical Results. *The Journal of Chemical Physics, 2*(11), 756-759. doi:10.1063/1.1749391

Mulvaney, R., Wolff, E. W., & Oates, K. (1988). Sulfuric-Acid at Grain-Boundaries in Antarctic Ice. *Nature, 331*(6153), 247-249.

Ondrušková, G., Krausko, J., Stern, J. N., Hauptmann, A., Loerting, T., & Heger, D. (2018). Distinct Speciation of Naphthalene Vapor Deposited on Ice Surfaces at 253 or 77 K: Formation of Submicrometer-Sized Crystals or an Amorphous Layer. *The Journal of Physical Chemistry C, 122*(22), 11945-11953. doi:10.1021/acs.jpcc.8b03972

Wettlaufer, J. S., & Worster, M. G. (2006). PREMELTING DYNAMICS. *Annual Review of Fluid Mechanics, 38*(1), 427-452. doi:10.1146/annurev.fluid.37.061903.175758

---

## Author Response (AR1)

**Reviewer #1 comments**

The manuscript by Hullar et al presents new experimental result on the photochemistry in snow. This is a controversial topic and more data to assess the role of snow as medium hosting chemistry that impacts atmospheric chemistry is needed. Therefore the manuscript's topic is fully within the scope of ACP. The manuscript is well written, conclusions are clearly supported by data and the argumentation is clear to follow. All relevant work is cited and addressed. I support publication in ACP after addressing some minor points:

The major point of the minor comments is a question: Do you have an explanation why the quantum yield are larger, and the photo decay more efficient, at the interface? If I'm not mistaken, dimethoxybenzene are well known to form triplet states upon irradiation in aqueous solution. Could this enhancement be a consequence of higher local concentrations and thus an increase in the secondary chemistry such as reduction induced by intermolecular electron transfer from such triplets? If so, I think it would be worth to mention, or even speculate on, this as it opens the door to a wider relevance. Triplets play a wide role in chemistry at the air ice interface for example in HONO formation (Bartels-Rausch, Atmospheric Environment, 2010)

We thank the reviewer for their comments. We do not have a preferred explanation for the apparent increase in quantum yield, although we have added some discussion of this in the revised manuscript. We deposited small amounts of dimethoxybenzenes (DMOBs) to the ice surface in an attempt to reduce areas of high local concentration, but it is possible the photodegradation rate constant enhancements are due to higher local concentrations increasing secondary chemistry. It is also possible that structural changes to the DMOB molecules on the ice surface could change the quantum yield. As discussed in our manuscript, our modeling suggests small geometric changes in the phenyl ring cause the modeled absorbance shifts; the change in the carbon atom positions might also change the quantum yield. An additional possibility is a reduced cage effect at the air-ice interface; this idea has been proposed previously (Meusinger et al. 2014) to explain observed enhancements in nitrate ion quantum yields. We have added text to the conclusion discussing the various reasons why quantum yields might be larger, and have included additional information about DMOB photochemistry, including formation of triplet states.

**Lines 16: «or in quasi-liquid layers (QLLs) at the air-ice interface, where they can readily exchange with the firn air.»**

Photochemistry could also occur at the air-ice interface at too cold temperature for QLL formation. Interestingly, some chemicals have been shown to form solvation shells at the interface (Bartels-Rausch, Earth and Space Chemistry, 2017), maybe this is even more relevant for photochemistry? Additionally, I might add that internal LLR could also be in exchange with air if located in grain boundaries as diffusion of any products from there is fast.

We thank the reviewer for the reference and information, which we have incorporated into our manuscript.

**Abstract: Could you detail a little more on the impact for atmospheric chemistry? The abstract remains a little bit too much on the technical result level for my taste. Please open up and explicitly mention the significance of this work.**

We revised the abstract to explicitly discuss the possible significance of our work.

**Lines 52-62: Leaves the reader wondering. Are reasons known? Do you have a suggestion? Are concentrations comparable? You mention high concentrations as a problem when determining absorption spectra, later in the manuscript. Could you exclude and/or discuss how the studies differ in this respect.**

While most of the cited studies present concentrations for aqueous solution and LLR samples, concentrations of test compounds at the air-ice interface were reported for only a fraction of the studies. Therefore, we do not have sufficient basis to address how concentration might explain the differences between photodegradation rate constants in the various studies.

**line 91: Does this paragraph only apply to solar, or also lab light sources?**

The first sentence here applies primarily to natural sunlight, but often to simulated sunlight as well. We have revised the paragraph.

**Line 107: photochemical reaction instead of loss?**

We have made the suggested change, and thank the reviewer for clarifying this point.

**Line 114: Any environmental relevance or these compounds that is worth highlighting? Please sketch structure and photo-chemical pathways, or refer to Figure 1 already here.**

DMOBs can be emitted into the environment by biomass burning; we have added this information to the paper. We chose these compounds as proxies for organic compounds in the cryosphere, rather than due to specific interest in these particular materials. We have modified the text to include reference to Figure 1 for DMOB chemical structures, and clarify the use of these compounds as model compounds to study photodegradation of aromatic organic compounds in the environment. Because we did not test our reacted solutions to determine photoproducts, any attempt to identify photochemical pathways would be highly speculative on our part, and of limited usefulness to the reader. We are not aware of any specific studies of the photochemical degradation pathways of these model compounds; however, we have added available information about observed reactions from DMOB photo-excited states.

**Line 163, I think the vapor pressures are also of high interest in this context.**

We thank the reviewer for the suggestion and have added vapor pressures to the paper.

**Line 271. Personally, I think it is odd to start the result section with data in the S1. That privilege should be reserved to data that makes it to a figure in the manuscript. Either make it a figure or start with current Figure 1.**

We understand the reviewer's concern. In order to provide fundamental data for review and transparency, we have chosen to include all the experimental data as supplemental figures, beginning with figure S1. We do not believe they should be included in the main paper since these figures are less important than Figure 1, which summarizes the experimental findings. In response to other comments, we have consolidated Sections 3.1 and 3.2, so although S1 through S12 are still presented before Figure 1, the gap is considerably smaller. While we agree that reference to a supplemental figure before a primary figure may be unexpected, we have seen other papers employing a similar approach and believe it is warranted in this case.

**Line 277: illuminated samples often show considerably greater loss, presumably due to photodegradation,**

**«presumably»...what else could it be? Why so careful?**

We agree that photodegradation is the most likely explanation for the results and have removed "presumably". We chose our initial wording to acknowledge the possibility of other processes (e.g., evaporation), but agree with the reviewer that it is overly cautious.

**Line 309: What is AB?**

"AB" indicates that the average rate constant for this sample treatment is statistically indistinguishable from the average rate constants of both groups A and B. Essentially, the uncertainty in these samples is so large that it cannot be differentiated from either group A or group B sample treatments, although it is different than group C. We have revised the Figure 1 caption to clarify this point.

**311. This is indeed a repetition from lines 271 -289. I suggest to delete 271 -289.**

We thank the reviewer for noting this opportunity to shorten the manuscript. We have combined the two sections and deleted significant amounts of text.

**324: Why is that. Is it because you had a larger surface to volume ratio in snow and thus higher concentration for analysis? What was the surface to ice volume ratio, anyway? Or, did you have more ideal flow conditions to deposit the chemical? How much did you deposit?**

Yes, the surface area to volume ratio of our snow samples is considerably larger than our frozen water ice samples. We did not measure the surface area to ice volume ratio; however, for a similar snow-making machine (Schleef et al. 2014), the snow specific surface area (SSA) was 600 cm2/cm3. Visual inspection of our snow revealed similar crystal types to those produced in the Schleef machine, and we expect our snow to have a similar SSA. In contrast, our VD to ice samples had an SSA of approximately 0.3 cm2/cm3. Amount of material deposited (as

determined by melted concentration) varied by compound and deposition method. For VD to snow samples, the range for all three isomers was  $0.1-13 \mu$ M, while for VD to ice it was  $0.4-6 \mu$ M; amounts deposited to snow were generally lower than to the ice. For each illumination experiment, we have added the initial DMOB concentration to the graph title for Supplemental Figures S1-S12, and modified the Figure caption accordingly. As the reviewer noted, we also believe we had more ideal flow conditions for chemical deposition to the snow than to ice surfaces. Perhaps more importantly, the deposited material for snow samples was spread across a surface area roughly three orders of magnitude larger than for ice samples, reducing the likelihood of aggregation. Aggregation could cause various effects, including different photodegradation rates due to change in reaction environment or increases in secondary chemical reactions. We have modified the text to discuss the possible reasons for better reproducibility when using nature-identical snow.

**Discussion & Results in general: could you discuss concentration dependence – and rule out secondary photochemical reactions.**

Unfortunately we did not collect data suitable for assessing the concentration dependence of DMOB photodegradation. We did not find any correlation between concentration and rate constants, but did not carry out experiments specifically to test this possibility. If secondary reactions were important – e.g., an excited state DMOB\* reacts with a neighboring ground state DMOB, leading to loss of one or both molecules – we would expect greater rate constants for loss in the VD-to-ice experiments compared to VD-to-snow. This is because surface concentrations of DMOB in the ice experiments were roughly 1000 times greater than in the snow, based on the relative specific surface areas of the substrates. But we do not see a greater loss on ice. Thus we believe our results are likely explained by direct photochemical degradation, but we cannot rule out secondary photochemical reactions.

**Comments from Dominik Heger, Reviewer #2**

The article assesses experimentally the rates of disappearance of three variously substituted dimethoxybenzenes in aqueous solutions, frozen in a freezer, liquid nitrogen, and deposited from the vapor phase onto the surfaces of ice and snow. 1,2- and 1,3 -DMOBs do not react in either a liquid or a frozen aqueous solution because they do not absorb applied light; 1,4-DMOB, by contrast, absorbs and reacts. Importantly, the vapordeposited compounds react in cases of all the isomers. Independently of the experiments, the absorption spectra of the compounds on the ice surface are calculated. The topic is appropriate for atmospheric chemistry and physics and features substantial novelty, as these compounds have not been photolyzed in/on ice to date. The research embodies an extension of the authors' previous papers on different compound (Hullar et al., 2020). I appreciate the fair effort of the authors to contribute towards expanding the knowledge of the photochemical fate of compounds in the frozen state, which is indispensable in describing the fate of the compounds in the cryosphere. The article presents a substantial amount of data. Regrettably, I have certain reservations concerning the quality of the data, correctness of the performed actinometry, and proper mathematical treatment of the kinetics and interpretation. Moreover, I have identified a number of nomenclature issues

that are inconsistent with their common use. I suggest that these issues be addressed by the authors prior to the acceptance of the manuscript.

**Conceptual problems:**

1. Equation 1 treats the heterogenous mixture as homogeneous. There is abundant literature to support the heterogeneity of the compounds in the frozen sample (Bartels-Rausch et al., 2014) (Heger et al., 2007). Generally, at least three environments are available: a) surface; b) veins and pools; c) crystals in which the QY (quantum yields) can vary. Regarding the formation of the crystals, the freeze-concentration effect and solubility should be discussed. The photochemistry in the crystals can yield photoproducts different from those in solutions. The authors, however, discuss only the concentrations of the reactants. Could they comment on the photoproducts as well? The heterogeneity of the reacting mixture can result in a non-monoexponential disappearance of the reactants; therefore, the adherence of the experimental data to the supposed monoexponential decay should be shown.

Equation 1 is for any given reservoir, as we now clarify in the text, and does not assume homogeneity across reservoirs.

Our sample preparation methods likely result in some heterogeneity, but we disagree with the assertion that our samples should be described as "heterogeneous". Each sample preparation method is designed to place solutes primarily in one specific compartment, and we use low concentrations to minimize crystallization or aggregation. In particular, in our vapor-depositedto-snow samples, solutes should be nearly completely at the air-ice interface, as there is insufficient time for snow metamorphism to occur and samples to be encapsulated in LLRs. Considering aggregation or precipitation of chemicals into crystals at the air-ice interface in this sample type, we have estimated surface coverage of DMOB molecules in the VD-to-ice as between 1.7 and 110 monolayers, while in the VD-to-snow samples the coverage is between 0.00036 and 0.047 monolayers. Even at our highest snow concentration, less than 5% of the available snow surface area would be covered with test compound. While it may be possible some molecules could aggregate and crystallize, it seems much more likely that the test compound remains solvated as individual molecules in the surface QLL. Bartels-Rausch et al. (2017) recently studied adsorption of formic acid onto the air-ice interface and reported molecules were present in solvation shells and restricted to the first few layers of the QLL; test compound surface coverage was approximately one monolayer, much higher than in our snow samples. Crystal formation may be more likely in LLRs formed by freezing solutions, where the freeze concentration effect may promote precipitation as DMOBs are concentrated above their solubility limit. However, some solute will remain in the LLRs.

We do not have information about the photoproducts from our reactions, as we did not attempt to identify the photoproducts.

Heterogeneous mixtures could result in non-monoexponential loss of reactants. To assess how closely our experimental results to monoexponential decay and allow other readers to evaluate our findings, we have included graphs of all illumination experiments in the supplemental

materials (Figures S1-S12). Although some of our data is noisy, we see no evidence of polyexponential decay. While we cannot rule out that our samples are somewhat heterogeneous, the arguments above, combined with our experimental findings, show that our results are attributable to monoexponential decay of test compound in largely homogeneous reservoirs.

**2. The authors should expose that the increased disappearance rates for the VD (vapor deposited) compounds are not due to merely the lower concentration in the irradiated samples. For this purpose, the absolute concentrations of the compounds should be provided.**

We thank the reviewer for this suggestion. For each illumination experiment, we have added the initial DMOB concentration to the graph title of Supplemental Figures S1-S12. We have also modified the Figure caption to explain how the initial DMOB concentration was determined. We see no apparent impact of concentration on photodecay rate constants.

**3. To calculate the QY, the knowledge of the absorbed number of photons is required; I do not assume this is reliably estimable from the current data (for the heterogenous samples and with a large uncertainty of the current disappearance rate constants and absorbance). Thus, I feel that most of the discussion is markedly speculative and should be announce in this way.**

We have clearly stated the assumptions we have used to estimate quantum yields. While we agree that our quantum yields contain considerable uncertainties, and discuss these in the text, the calculations are based on laboratory measurements and general photochemical principles. Therefore, we do not believe they should be characterized as "speculative", any more so than the underlying data are. We have revised the text in several places to reiterate that the quantum yields are estimated from calculations.

4. The absorption spectra were simulated for ice surface. How can the authors ensure that they are correct? Does the applied method reproduce solvatochromic shifts in a solution for some known compounds? Can the simulation be done for an FCS (freeze concentrated solution)? Concerning further issues associated with the simulation of the spectra: I suspect the formation of crystals on the ice surface by vapor deposition (Ondrušková et al., 2018) and frozen (amorphous) solution in the veins by LN freezing. (Ondrušková et al., 2020) Therefore, I think that the simulation should attempt to distinguish the FCS, ice-air interfaces and crystals of the compounds. In other words, even though the simulation could correctly predict interaction of compounds with the ice surface, that does not need to be the main species in the prepared sample. The authors are invited to address this issue.

Whereas it is not possible to have a direct benchmark of the solvatochromic shift for molecules adsorbed on ice at environmental concentrations, we have tested our approach on phenol in the gas phase and in water at low concentration at room temperature. Our multi-model method reproduces the solvatochromic shift of  $\sim$ 9 nm for phenol from gas phase (261 nm) to aqueous solution (270 nm). Also the line-shape of the aqueous spectrum is predicted accurately (Bononi et al. 2020). We have added mention of these results in the revised manuscript.

Regarding potential modeling of FCS or crystals, the reviewer makes an excellent point and modeling could in principle address these important questions. However, modeling absorbance of solute aggregation in freeze-concentrated solution or in crystals formed at the air-ice interface is beyond the scope of our work and would require substantial human and computational resources and time. For example, in concentrated solution we would need to simulate the direct interaction of two or more solute molecules. The calculations presented in our submitted manuscript required about 300,000 CPU hours per molecule. Given that the computational cost of the direct calculation of the spectra scales like the number of atoms (or solute molecules) to the fourth power, the computational cost would increase to 4.8 million CPU hours per molecule. This is possible on a high performance computing facility, but it would require more than a year to get the computer time and carry out the calculations.

**5. As the employed actinometer, *o*-nitrobenzaldehyde, absorbs and photoreacts up to 366 nm and 404 nm (Leighton et al., 1934), the correction for the DMOBs is very imprecise and possibly inapplicable. I propose that the absorption spectra of the studied compound be plotted together with *o*-nitrobenzaldehyde to show the spectral overlap (or the lack of it). Monochromatic light (filtered low pressure mercury lamp) would be preferable for the actinometry (besides the polar sun experiments).**

We disagree that the use of *o*-nitrobenzaldehyde (abbreviated as 2NB in our work) as a chemical actinometer is inaccurate or inapplicable. As described in our previous paper and referenced in this work, we used *j*2NB measurements - along with separate measurements of the wavelength-dependent relative photon flux - to calculate the wavelength-specific photon flux in each of our sample preparations. Because these calculations gave photon fluxes at each wavelength, it is not critical that the 2NB absorbance range match that of each of the DMOBs. In fact, the wider absorbance range of 2NB allows estimation of experimental photon fluxes throughout the span of DMOB absorbances. Artificial sunlight is better than monochromatic illumination in these experiments because the former simulates the large increase in photon flux with increasing wavelength, which is not the case with monochromatic illumination.

**6. The aggregation of compounds in nature is reported in ice, e.g., (Mulvaney et al., 1988). I agree that both the separated and the aggregated compound can photoreact. It would be worth discussing under which natural circumstance the compounds do and do not aggregate. As a matter of fact, I cannot find in the article a proof that the compounds are not aggregated under the current experimental conditions.**

We agree that the compounds can aggregate in nature and in laboratory settings. We have developed our experimental methods, including the use of nature-identical snow and low chemical concentrations, to minimize the likelihood of aggregation. However, we cannot prove our chemical compounds are present only as individual molecules on the ice surface, rather than aggregated areas of pure or concentrated material, and are not aware of any method to rule it out. Beside concentration, it is unclear what factors influence aggregation in the natural environment. We have, however, added text in the manuscript discussing the possibility of aggregation.

**Suggestions:**

**1. The mechanisms, photoproducts, and rate constants for the photochemically induced reactions of DMOBs should be reviewed to allow a mechanistical explanation of the proposed changes.**

We agree that understanding the mechanisms and pathways of DMOB photodegradation would provide interesting insights. Unfortunately, there is only limited information available exploring these processes, and our laboratory is not equipped to identify and quantitate the photoproducts. We have expanded the introduction to more thoroughly discuss available literature on DMOB photodegradation.

2. The introduction and results suffer from permanent referencing to previous research of the authors, to such an extent that the present manuscript is not comprehensible without simultaneous reading of the articles published in the past. In this context, I strongly recommend that a full description of the sample preparation be included in particular. For a reader, it is generally inconvenient to have to refer to the authors' previous publications to learn how each sample was prepared.

We understand the reviewer's concern, and thank him for his patience in looking at past work. We have tried to strike a balance between including enough text for our approach to be clear, while not repeating previously reported details that are of interest only to those highly specialized in the field. We have made some slight modifications to the section in question for clarity and flow. We have also added a new section to the Supplemental Materials providing a more comprehensive description of our sample preparation methods, which should reduce or eliminate the need to reference past works.

**Formal and nomenclature problems:**

1. QLL: the term is not used properly, e.g., lines 44-45, Table 1, and throughout. QLL (synonyms disordered interface, surface melting...) forms on the surfaces of pure water ice. At a moment of inserting impurities in, the layer is not QLL but a brine layer – that should be recognized in the manuscript. In the manuscript, the QLL is sometimes also confused to specify the ice-air surface. QLL forms in any surface of ice crystals, not just on the ice-air interface but also at grain-boundaries (Wettlaufer et al., 2006); this use of the term is also in (Barret et al., 2011) for example. A quasi brine layer was introduced to describe the brine at subeutectic temperatures (Cho et al., 2002). I strongly recommend that these distinct terms be kept in separate uses to avoid further confusion.

The use of "QLL", "LLR", "brine layer", and other terms in the literature has been a source of ongoing discussion and disagreement in the community. We do not feel the term "QLL" has been standardized to the meaning given by the reviewer and it has been suggested that some measurements of the QLL thickness at ice surfaces were influenced by impurities. We note the use of QLL to represent a surface disordered region containing solutes by Kahan, Reid, and Donaldson (2007), and Boxe and Saiz-Lopez (2008), among others. The use of two different terms (QLL and brine layer) to represent a change as small as the addition of a single molecule seems an arbitrary distinction to us, and does not accurately represent the range of chemical concentrations present at the air-ice interface in the natural environment. Given the use of

various terms in the literature, we continue using QLL to represent the disordered region at the air-ice interface, whether or not it contains solutes. We have discussed and defined our use of these terms in the introduction, and believe that while a reader may disagree with our definition, our usage of the various terms are clear in the text. To attempt to eliminate any confusion and acknowledge the variation in use of "QLL" in the literature, we have modified the text of our introduction.

2. I would like to ask the authors to adhere to the correct nomenclature regarding the disappearance of the chemical compounds. Firstly, the "photochemical reaction rate constant" (line 122) is not defined. A photochemical reaction starts with the absorption of light and is followed by some relaxations like intersystem crossing, internal conversion, fluorescence, and reactions from the excited state - the individual processes can be characterized by rate constants and quantum yields. The rate constants are typically fast  $(k=10e7-10e3 s^{-1})$ , measured by techniques applying pulsed lasers (Givens et al., 2008; Klán et al., 2009). Such rate constants are, however, not measured here. I feel strong urge to create the nomenclature that avoids any possible confusion between these rate constants and the apparent rates of compounds disappearance. Thus, statements such as that on line 106 "... a faster reaction rate,..." (and many others) should be avoided, as the reaction rate implicitly describes the rate of some reaction step or overall reaction, which is not being determined here. In reality, the reaction rates can be altered by the freezing; thus, statements like that on line 106 can be interpreted in a other way than that intended by the authors. What the authors are describing (eq. 1) should be, in my view, called the apparent rate of compound disappearance. The adjective "apparent" or an applicable synonym would clearly indicate that no rate of chemical reaction or of an elementary step is discussed. The measured apparent rate of compound disappearance depends of the photon flux, distance from the lamp, filters applied, ... whereas the rate of the chemical reaction does not.

We are unclear why the reviewer says that the rate of the "overall reaction...is not being determined here". We experimentally measured the loss of compound due to photochemical degradation, and based on that data, determined the first-order rate constant of photochemical loss. The loss of chromophore is a result of very rapid photochemical steps (e.g., excitation to a singlet state, likely intersystem crossing to a triplet state, then decomposition of the excited state molecule), but the overall process is commonly described with a first-order photochemical rate constant for decay, as we have done here. This does not diminish the value of our findings - we have determined the overall rate of a chemical reaction under controlled and specified conditions. Yes, the rate constants of the reactions depend on the photon flux, but we also used our measurements and modeling to determine the two fundamental properties - molar absorption coefficients and quantum yield - that control the photochemistry of each molecule, independent of photon flux. While it is interesting to understand the nanosecond time scale processes that lead to chromophore loss, we have determined the most important quantities from an environmental perspective. Finally, we disagree that use of the word "apparent" would clarify what we have determined; to the contrary, "apparent" suggests photodegradation was not happening or was a secondary process, and that other processes can better explain the results, which we do not believe to be true. To attempt to avoid any possible confusion with other types

rate constants, we have changed our terminology to emphasize we are determining the "photochemical reaction rate constant for loss".

**3. Molar absorptivity is not correct term. I suppose the author means "molar absorption coefficient" and should use this term throughout.**

(https://goldbook.iupac.org/terms/view/M03972)

While we have happily used the term "molar absorptivity" for several decades, we thank the reviewer for alerting us to the IUPAC preference. We have changed this throughout the paper.

**4. The rate constant for the light absorption is a term without a physical meaning in a discussed context. Light is absorbed instantly (in order of femtoseconds or less).**

This first-order rate constant for light absorption expresses the rate of light absorption per molecule per second under our experimental conditions. It is a function of the photon flux in our system and the strength of light absorption by the molecule. We have modified the text to clarify the term.

**Data collection problems**

Importantly, the presented data mostly report insufficient sampling and reproducibility, together with inadequate irradiation times or a too weak light source. For example, Figure S2a shows an increasing concentration of the compound in the dark experiment. How is this explained? I suggest statistical testing if the concentration after the preset time (at the end of the irradiation) significantly differs from that at the beginning. Besides the data in Figures S6f, S7a-c, 10, 11b-c, I do not think the statistical testing would reveal the influence of irradiation (or time in general). In the negative case, the data should not be used in further analysis, and the experimental design should be improved.

Example Figure1c.: The conversion in 1,500 min ranges from ca 0.5 to 2 %. Such a small conversion with this large spread does not allow reliable estimate of the apparent rate constant. If the uncertainty of the slope were calculated, this fact would be apparent.

The data in Figure 1 are presented in such a manner that there does not seem to be any difference in the rate of photodegradation in first three preparation approaches. As a matter of fact, a closer look at Figures S1-12 indicates that under the performed experimental conditions the difference cannot have been detected. This fact must be emphasized in the manuscript. The post hoc Tukey-Kramer test does not reveal the issue. The performed analysis (discussed in the text) corroborates the inconclusive experiments, as most of the outputs are not significant and are highly scattered. This makes the appropriateness of publishing these data questionable.

The negative apparent rates of photochemical degradation should not be reported and should warn the authors about their experimental design.

At the same time, considering my own experience in the preparation of frozen samples, I know that these samples tend to produce heterogenous outputs; therefore, I really value the

authors' work. Only, I press on not to conclude from the data the outputs that they do not offer. Still, the difference in the vapor-deposited samples is apparent and worth publishing, on the condition that the absolute concentrations are shown.

We appreciate the reviewer has taken time to review our experimental results in detail. We chose three compounds which absorb little to no light in solution to see whether their absorbance and photochemistry are enhanced in, and on, ice. Thus we see little to no direct photochemistry in solution and, as it turns out, under some of our ice conditions, leading to very noisy data. As the reviewer mentions from his own work, noisy output is quite common when performing photochemistry experiments in frozen water systems. We agree, as the reviewer stated, that the "difference in the vapor-deposited samples is apparent and worth publishing", and feel this is one of the major points of the research presented here. As discussed in greater detail in other comments, we have added the absolute concentration information requested.

We agree that the DMOB loss observed in Supplemental Figure 1c (among others) is very small. The 95% confidence interval for the photodegradation rate constant, calculated from the standard error of the line, is  $3.6 \times 10^{-6}$  min-1, or about a third of the value of the measured rate constant. Whether or not this represents a "reliable" measure of the true rate constant is somewhat a matter of opinion, but we note the value determined from this experiment is statistically greater than zero and therefore provides some estimate of the rate constant.

We disagree with the reviewer's assertion that only Supplemental Figures S6f, S7a-c, S10, and S11a-c show the influence of irradiation. In fact, we performed statistical testing immediately after every experiment and used that data to refine illumination times and other factors driving data quality. Only 13 of 51 illumination experiments showed illuminated sample rate constants indistinguishable from zero: S2a-c (1,2-DMOB in frozen solution); S3a, c, and d (1,2-DMOB in LN2 frozen samples); S4b (1,2-DMOB VD to ice surface); S5b and c (1,2-DMOB VD to snow); S6c and d (1,3-DMOB aqueous); and S11a and e (1,4-DMOB VD to ice surface). For the remaining 75% of the experiments, the influence of irradiation was clear and indicated by photodegradation rates statistically greater than zero.

While longer irradiation times or a brighter light source might provide better estimates of some of the slower photodegradation rates, our illumination source was designed to approximate natural polar sunlight, rather than precisely determine slow photodegradation rate constants. We believe the irradiation times used here are suitable to determine the slow photodegradation rates with sufficient accuracy to evaluate the relative enhancements seen at the air-ice interface.

Although some experimental data appear to show increasing concentration and therefore DMOB formation in dark samples, statistically these slopes are generally indistinguishable from zero. We did not find any reproducible evidence of DMOB formation in dark samples, and believe the cases of positive slopes are attributable to experimental noise. For illuminated samples, only two experiments showed apparent DMOB formation (S2c and S3c); in both cases, this "formation rate" was statistically indistinguishable from zero.

We agree that the apparent rates of photodegradation in the first three sample preparation methods are slow and the data frequently are noisy. To address this concern, we note in the text

those datasets where the average photodegradation rate constant is indistinguishable from zero; for the convenience of the reader, we have also updated Figure 1 to include this information. However, we do not feel it appropriate to reject this data from further consideration; datasets where the average photodegradation rate constant is indistinguishable from zero provide a valuable comparison to those where the rate constants are statistically greater than zero. The uncertainties of these comparisons are clearly presented in Table 1, as well as Figure 1. Further, the Tukey-Kramer test provides a statistically rigorous method for differentiating datasets, and includes their variability.

We thank the reviewer for noting the value of the vapor-deposited samples, and agree they are worthy of publication. Despite the occasionally noisy data (and subsequent uncertainties), we believe the comparison of the various sample preparation methods puts the vapor-deposited samples into context, and should be part of the paper.

**Data treatment problems**

A major question rests in whether the observed rate of the compound disappearance, under steady irradiation, will follow a monoexponential decay. To respond to this issue, there are a number of reasons that indicate why it will not: For example, if the ice creates a non-homogeneous environment where compounds undergo degradation with unequal quantum yields or if the photoproduct would absorb. Importantly, the authors should show the data and analyze their deviations (residuals) from the monoexponential decay (without the logarithmic linearization).

Linearization of the kinetics was required before the availability of computers. Today, there is no basis for the method. Moreover, the exponential fit provides proper weighting of each of the points, whereas linearized one does not. Fitting the exponential decay curve and plotting the data in this way would allow the authors to determine more easily if the fit makes sense and if the sample was exposed to the light long enough.

Our data are noisy, but they are best explained by a first-order decay process. We included plots of our experimental data (Supplemental Figures S1-S12) so the reader could review the data independently and draw their own conclusion. Linearizing first-order kinetics data remains a common method to analyze and present kinetic findings. It has the advantage of being computationally simple, easy to understand, and intuitive to present, which may explain its ongoing popularity and common usage. It also clearly shows deviations from monoexponential decay. While we understand linearization adds some statistical biases, we believe that is a very small source of error in our experimental datasets, especially compared to the noise in our concentration data. Linearizing our data allows us to accomplish our main objectives, which are to demonstrate the data can be described using first-order kinetics and to assess the differences in photodegradation rate constants for loss determined using various sample preparation methods. While linearization may introduce very small statistical biases, we have found the benefits to data quality assessment and overall understanding justify its use. Perhaps most importantly, we have no reason to think the small statistical advantage of an exponential fit will contradict our determination of first-order decay in our photodegradation experiments.

**Sample preparation:**

**Importantly, the absolute concentrations of the compounds in the VD (vapor deposited) samples should be given.**

We have modified the supplemental figures to include initial chemical concentration, as described in more detail above.

Further, the exact temperature of the vapor deposition may play a role, too: The fluorescence spectroscopy of naphthalene showed that an amorphous layer forms at 77 K, whereas microcrystals emerge at 253 K. (Ondrušková et al., 2018) The monolayer was never detected. I suspect the microcrystals formation under the here reported "laboratory freezer" experimental temperature.

We cannot rule out DMOB precipitation and subsequent crystal formation, either in the vapor deposition or laboratory freezer samples. We note, however, that such precipitation would likely be incomplete, and some compound should remain dissolved in the LLR regions. Given that previous studies frequently find photodegradation rate constants in LLRs similar to that in aqueous solution, we would tentatively conclude our results here represent photodegradation rate constants in LLRs, not in crystals. However, this is not a strong finding, and would require more research to substantiate.

**Particular comments:**

**44-46 The sentence is misleading for above mentioned reasons.**

As discussed in more detail above, we have changed this sentence to acknowledge the variation in usage of "QLL" and more clearly define how we use it in our manuscript.

**96 The cited publication often observes the bathochromic shifts in both frozen and VD samples. Comparison of these differences may be important for the current study.**

Our text here does note that shifts were found in both LLRs (frozen samples) and at the air-ice interface. Unfortunately, only two of the cited studies measured absorbance shifts in both LLRs and QLLs, providing no strong evidence for broader conclusions. Due to available computational resources, we did not model DMOB absorbance in LLRs, but only QLLs. Unfortunately, this means we have no basis from which to determine the absorbance spectrum of each DMOB in LRRs, and have therefore assumed them equivalent to their water absorption spectra.

**98-102**

Importantly, there is a literature report on 100 nm shift in the absorbance upon freezing. (Heger et al., 2007) As a matter of fact, the aggregation on the ice surfaces in the experiments does always need to be a disadvantage: the same processes are to be expected to occur in the nature during the metamorphosis processes.

We thank the reviewer for this reference, and have revised the text to include this information. We share the reviewer's perspective that aggregation is a natural and interesting process, and believe more work needs to be done to understand absorbance shifts and photochemical degradation under a variety of conditions. Here, our work is primarily motivated to understand the nature of photochemical reactivity at the air-ice interface under natural conditions. Therefore, our research has tried to focus on the low concentrations, and subsequent less likelihood for aggregation, present at the air-ice interface of polar snow and ice.

**124 "normalizing each to the measured photon flux.." I am worried, that the photon flux is measured at the different wavelengths, thus is not much applicable.**

The purpose of normalizing photodegradation rate constants to measured photon flux in each sample type, using photolysis of 2-nitrobenzaldehyde (2NB), was to allow a correction for variations in local photon flux in each sample preparation type. As could be expected, and documented in our manuscript, the local photon flux in snow is considerably less than in aqueous solution; if left uncorrected, the photodegradation rate constant would appear to be slower than it really is. While we agree that 2NB does not share the same light absorption spectrum as our test compounds, it would only matter if the samples were significantly changing the spectrum of the experimental light passing through them. In our work, the experimentally measured 2NB photodecay rate constants are proportional to the total amount of light in the sample, allowing normalization of our experimental photodegradation results across sample preparation methods.

**168 How was the homogenous irradiation guaranteed? Were the samples rotated? Please, provide some detail, how much variance can be the photon flux and how much variability comes from the sampling and chemical analysis.**

Although we attempted to diffuse our light source as much as was practical, sample irradiation was not perfectly homogeneous. As described in the text and tabulated in Supplemental Table S1, the illumination intensity varied across the various sample positions of the sample holder, but we corrected for this in our calculations. The samples were not rotated. Supplemental Table S3 contains averages and standard deviations for both the photodegradation rate constants ( $j_{DMOB}$ ) and the 2NB photodecay rate constants ( $j_{2NB}$ ). In general, most of the experimental variability is attributable to the measurement of DMOB photodegradation rate constants, which would include sampling and analysis. For most sample preparations, the contribution of photon flux variation (as measured by 2NB photolysis) is a minor contributor.

**204 Why should the dark loss follow first order disappearance? And does it?**

DMOB loss in dark (nonilluminated) samples was generally small, so it's difficult to firmly conclude it follows first-order disappearance. Within these limits, we observed that dark loss can be estimated as a first-order loss process. We expect that loss of DMOB from dark samples would be due to volatilization or reactions with other contaminants, both first-order or pseudo-first-order processes.

312 "normalized rate constant is statistically indistinguishable from zero" This is important. This is saying that the data are too noisy to draw any conclusion from. I think the authors should stress this fact. Still, the possible conclusion is that VD samples react more readily then the aqueous or frozen.

While we agree the data are noisy, we disagree that no conclusion can be drawn as a result. For example, the Tukey-Kramer test shows that the sample population for 1,2-DMOB aqueous samples is different than the VD-to-snow sample population. While we would certainly have preferred if our sample results were less noisy and our confidence intervals narrower, we believe we have presented our results adequately and drawn appropriate conclusions.

**375 The experimental spectra of molar absorption coefficients reported in SI are valuable contributions. There will be even more valuable, if provided with standard errors of means, which should be available to the authors. Please, provide them.**

We thank the reviewer for the suggestion and have added this information.

**488-490 The absorption properties in FCS is possibly more altered then on the ice surface due to more intensive interactions in that environment.**

Conceptually we agree with the reviewer: the high solute concentrations in a LLR could lead to an environment promoting greater interactions between solutes. However, we note that while some previous studies found absorbance shifts for compounds in LLRs, others did not. Given the uncertainty around this topic, we felt it most appropriate to make the conservative assumption that the LLR absorbance spectra were the same as in aqueous solution. We have modified the text to clarify our assumption.

**491 A reference for the QY is missing.**

The result presented here is a calculated QY based on our experiments and modeled absorbance shift. We apologize for the confusion, and have modified the text to clarify this point.

**539 ... in snow or ice" Could the authors elaborate what changes do they expect on these two environments and why?**

We have revised the text to more thoroughly discuss possible lifetime changes.

**608 Gaussian curve fitting. Before the fit, data should be converted to scale with energy. (Antonov, 1997) More than two transitions are apparent – as discussed in the text (611) however only one curve is plotted in the Figure S19.**

We apologize for the confusion here, and have revised the text to more clearly explain our process. We plotted one Gaussian curve for 1,2-DMOB and another Gaussian curve for 1,4-DMOB, for a total of two curves. Using the parameters for these curves (width, location, and height) we derived a single Gaussian curve to represent the absorbance spectrum of a single hypothetical model compound. We could have used other curve shapes, and chose a Gaussian because it was straightforward to manipulate mathematically and adequately conformed to the observed spectral shapes for the two DMOB.

**Reviewer 2 references**

Antonov, L. (1997). Drawbacks of the present standards for processing absorption spectra recorded linearly as a function of wavelength. *Trac-Trend. Anal. Chem., 16*(9), 536-543.

Barret, M., Domine, F., Houdier, S., Gallet, J. C., Weibring, P., Walega, J., Fried, A., & Richter, D. (2011). Formaldehyde in the Alaskan Arctic snowpack: Partitioning and physical processes involved in air-snow exchanges. *Journal of Geophysical Research-Atmospheres*, *116*. doi:D00r03 10.1029/2011jd016038

Bartels-Rausch, T., Jacobi, H. W., Kahan, T. F., Thomas, J. L., Thomson, E. S., Abbatt, J. P. D., Ammann, M., Blackford, J. R., Bluhm, H., Boxe, C., Domine, F., Frey, M. M., Gladich, I., Guzmán, M. I., Heger, D., Huthwelker, T., Klán, P., Kuhs, W. F., Kuo, M. H., Maus, S., Moussa, S. G., McNeill, V. F., Newberg, J. T., Pettersson, J. B. C., Roeselová, M., & Sodeau, J. R. (2014). A review of air-ice chemical and physical interactions (AICI): liquids, quasi-liquids, and solids in snow. *Atmos. Chem. Phys.*, *14*(3), 1587-1633. doi:10.5194/acp-14-1587-2014

Givens, R. S., Heger, D., Hellrung, B., Kamdzhilov, Y., Mac, M., Conrad, P. G., II, Cope, E., Lee, J. I., Mata-Segreda, J. F., Schowen, R. L., & Wirz, J. (2008). The Photo-Favorskii Reaction of p-Hydroxyphenacyl Compounds Is Initiated by Water-Assisted, Adiabatic Extrusion of a Triplet Biradical *Journal of the American Chemical Society*, *130*(11), 3307-3309.

Heger, D., & Klan, P. (2007). Interactions of organic molecules at grain boundaries in ice: A solvatochromic analysis. *Journal of Photochemistry and Photobiology a-Chemistry*, 187(2-3), 275-284. doi:10.1016/j.jphotochem.2006.10.012

Hullar, T., Bononi, F. C., Chen, Z., Magadia, D., Palmer, O., Tran, T., Rocca, D., Andreussi, O., Donadio, D., & Anastasio, C. (2020). Photodecay of guaiacol is faster in ice, and even more rapid on ice, than in aqueous solution. *Environ Sci Process Impacts*, *22*(8), 1666-1677. doi:10.1039/d0em00242a 6

Cho, H., Shepson, P. B., Barrie, L. A., Cowin, J. P., & Zaveri, R. (2002). NMR investigation of the quasi-brine layer in ice/brine mixtures. *Journal of Physical Chemistry B*, 106(43), 11226-11232.

Klán, P., & Wirz, J. (2009). *Photochemistry of Organic Compounds: From Concepts to Practice:* Wiley.

Leighton, P. A., & Lucy, F. A. (1934). The Photoisomerization of the o-Nitrobenzaldehydes I. Photochemical Results. *The Journal of Chemical Physics*, 2(11), 756-759. doi:10.1063/1.1749391

Mulvaney, R., Wolff, E. W., & Oates, K. (1988). Sulfuric-Acid at Grain-Boundaries in Antarctic Ice. *Nature, 331*(6153), 247-249.

Ondrušková, G., Krausko, J., Stern, J. N., Hauptmann, A., Loerting, T., & Heger, D. (2018). Distinct Speciation of Naphthalene Vapor Deposited on Ice Surfaces at 253 or 77 K: Formation of Submicrometer-Sized Crystals or an Amorphous Layer. *The Journal of Physical Chemistry C, 122*(22), 11945-11953. doi:10.1021/acs.jpcc.8b03972

Wettlaufer, J. S., & Worster, M. G. (2006). PREMELTING DYNAMICS. *Annual Review of Fluid Mechanics*, 38(1), 427-452. doi:10.1146/annurev.fluid.37.061903.175758

**Response references**

Bartels-Rausch, T., F. Orlando, X. R. Kong, L. Artiglia and M. Ammann: Experimental Evidence for the Formation of Solvation Shells by Soluble Species at a Nonuniform Air-Ice Interface, Acs Earth and Space Chemistry, 1(9), 572-579, doi: 10.1021/acsearthspacechem.7b00077, 2017.

Bononi, F. C., Z. K. Chen, D. Rocca, O. Andreussi, T. Hullar, C. Anastasio and D. Donadio: Bathochromic Shift in the UV-Visible Absorption Spectra of Phenols at Ice Surfaces: Insights from First-Principles Calculations, Journal of Physical Chemistry A, 124(44), 9288-9298, doi: 10.1021/acs.jpca.0c07038, 2020.

Boxe, C. S. and A. Saiz-Lopez: Multiphase modeling of nitrate photochemistry in the quasiliquid layer (QLL): implications for NOx release from the Arctic and coastal Antarctic snowpack, Atmospheric Chemistry and Physics, 8(16), 4855-4864, 2008.

Kahan, T. F., J. P. Reid and D. J. Donaldson: Spectroscopic probes of the quasi-liquid layer on ice, Journal of Physical Chemistry A, 111(43), 11006-11012, doi: 10.1021/jp0745510, 2007.

Meusinger, C., T. A. Berhanu, J. Erbland, J. Savarino and M. S. Johnson: Laboratory study of nitrate photolysis in Antarctic snow. I. Observed quantum yield, domain of photolysis, and secondary chemistry, Journal of Chemical Physics, 140(24), doi: 10.1063/1.4882898, 2014.

Schleef, S., M. Jaggi, H. Lowe and M. Schneebeli: An improved machine to produce natureidentical snow in the laboratory, Journal of Glaciology, 60(219), 94-102, doi: 10.3189/2014JoG13J118, 2014.